# Retro*: Optimizing LLMs for Reasoning-Intensive Document Retrieval

**Junwei Lan**[1,2,4*] **Jianlyu Chen**[1,2,4*] **Zheng Liu**[2,5†] **Chaofan Li**[2,3] **Siqi Bao**[6] **Defu Lian**[1,4†]

[1] University of Science and Technology of China    [2] Beijing Academy of Artificial Intelligence
[3] Beijing University of Posts and Telecommunications
[4] State Key Laboratory of Cognitive Intelligence    [5] Hong Kong Polytechnic University
[6] Hong Kong University of Science and Technology
`ljw13@mail.ustc.edu.cn`   `zhengliu1026@gmail.com`   `liandefu@ustc.edu.cn`

## Abstract

With the growing popularity of LLM agents and RAG, it has become increasingly important to retrieve documents that are essential for solving a task, even when their connection to the task is indirect or implicit. Addressing this problem requires fine-grained reasoning to accurately assess the relevance between the task and each candidate document. This capability, however, poses a significant challenge for existing IR techniques. Despite recent progress in reasoning-enhanced IR, existing approaches still face significant challenges in applicability, scalability, and efficiency. In this work, we propose **Retro***, a novel approach for reasoning-intensive document retrieval. Our method introduces a rubric-based relevance **scoring mechanism**, enabling the model to reason about the relationship between a task and a document based on explicitly defined criteria, whereby producing a fine-grained, interpretable relevance score. Retro* also supports **test-time scaling** by combining multiple reasoning trajectories via score integration, which produces more reliable relevance estimates. To optimize Retro*'s reasoning capabilities, we introduce a novel **reinforcement learning** algorithm tailored for its relevance scoring mechanism, which employs two composite rewards to fully exploit the trajectories of each training sample. Our experiments show that Retro* outperforms existing document retrieval methods with notable advantages, leading to **state-of-the-art** performance on the BRIGHT benchmark.

## 1 Introduction

Large language model (LLM) agents have become increasingly important for tackling complex tasks such as software engineering, mathematics, and scientific research (Chan et al., 2024; Jin et al., 2025; Wei et al., 2025; Phan et al., 2025). In these applications, retrieval-augmented generation (RAG) (Lewis et al., 2020; Gao et al., 2023) plays a crucial role, as access to external knowledge is often necessary to produce high-quality solutions. However, in many scenarios, retrieval models must identify useful documents, even when their connection to the task is indirect or implicit, which makes the retrieval process particularly challenging. For example, in software engineering, a retrieval model may need to locate programs that share similar design patterns with the target problem rather than matching exact code snippets (Jimenez et al., 2023). In mathematics, it might involve retrieving proofs derived from the same underlying theorem, even if they are expressed differently (Chen et al., 2023). Solving such tasks requires fine-grained reasoning to bridge subtle connections between the task and candidate documents. However, existing retrieval models are primarily designed to capture straightforward semantic relationships, such as matching question-answer pairs or identifying paraphrases (Lee et al., 2019; Karpukhin et al., 2020). Consequently, they often struggle with the complex reasoning required to uncover these deeper, more abstract connections.

Enhancing the reasoning capabilities of LLMs for reasoning-intensive tasks is a central focus of current research. One promising strategy is test-time scaling, which guides or reinforces LLMs to

---

*Core Contributors.
†Corresponding authors.

generate long-form thoughts before arriving at a final answer at inference time (Wei et al., 2022; Jaech et al., 2024; Guo et al., 2025). This approach also enables the LLMs to explore multiple reasoning paths, evaluate alternatives, and ultimately arrive at more accurate solutions for complex problems (Wang et al., 2022). Inspired by these advances, recent information retrieval (IR) research has begun to leverage the reasoning capabilities of LLMs for reasoning-intensive document retrieval tasks (Su et al., 2024; Niu et al., 2024; Zhuang et al., 2025; Weller et al., 2025). Current methods typically follow one of two paths: some directly prompt general-purpose LLMs to perform fine-grained relevance analysis, while others optimize these models with fine-tuning algorithms to elicit more structured and systematic reasoning behaviors. Despite recent progress, current approaches exhibit three key limitations in relevance-measuring functionality, test-time scalability, and parallelism:

- **Lack of relevance-measuring functionality**. Many RAG applications require a specific functionality: the direct and interpretable measurement of document relevance. However, existing methods primarily provide relative ranking orders, which cannot capture the absolute level of relevance needed by downstream tasks.

- **Inflexible test-time scalability**. Existing methods mainly focus on generating a single, long-form thought to reach an answer. However, they neglect the significant potential of exploring and integrating multiple reasoning paths to achieve more reliable performance.

- **Limited parallelism capability**. Existing methods, which are primarily listwise (Sun et al., 2023) or setwise (Zhuang et al., 2024), must sequentially process the entire candidate set to produce the final retrieval result. This inherently sequential design is prone to substantial latency, especially when handling a large group of candidate documents.

In this work, we propose **Retro\* (Retro-star)**, a novel LLM-based retrieval model designed for reasoning-intensive IR tasks. Distinct from existing approaches, Retro* is built on two key designs:

- **Rubric-based relevance scoring**. Retro* introduces a fine-grained set of relevance rubrics, which explicitly define the relevance scores and their interpretations. Based on these rubrics, Retro* performs pointwise reasoning on the relationship between a query and its candidate documents, producing concrete relevance scores with clear, interpretable meanings. This design enables direct measurement of relevance, rather than merely providing a relative ordering of documents.

- **Test-time scaling by score integration**. Building on its rubric-based scoring, Retro* supports test-time scaling by generating multiple trajectories for each query-document pair and integrating their individual scores based on score similarity, resulting in a more reliable and stable estimate of document relevance.

With these designs, Retro* can serve both relevance measuring and re-ranking applications. Furthermore, its architecture naturally supports both flexible test-time scaling and high parallelism, making it proficient at performing effective and efficient reasoning for complex retrieval problems.

To bring these designs to their full potential, we introduce a novel reinforcement learning (RL) algorithm with two composite rewards to further optimize Retro*'s capabilities. The composite rewards are designed to fully exploit the trajectories of every training sample during RL training. The **Intra-Document Reward** guides the policy model to assign accurate relevance scores for each individual document, whereas the **Inter-Document Reward** incentivizes the policy model to effectively discriminate the relevant document from an irrelevant one. To stabilize the training process and provide Retro* with an initial reasoning ability, we incorporate a warm-up supervised fine-tuning (SFT) stage. This stage not only equips the model with basic reasoning skills, but also shapes the model to generate concise and well-structured thoughts before the RL stage.

To evaluate the effectiveness of our training strategy and the overall performance of Retro* on reasoning-intensive IR tasks, we conduct experiments on BRIGHT (Su et al., 2024), a comprehensive benchmark encompassing 12 datasets across science, mathematics, and programming. Experimental results demonstrate that Retro* achieves significant improvements over strong baseline methods, with substantial performance gains from the proposed test-time scaling mechanism and reinforcement learning method. To facilitate future research in this area, all resources are released at `https://github.com/VectorSpaceLab/agentic-search/tree/main/Retro-star`.

```
Here is the relevance definition... RELEVANCE_PLACEHOLDER
Now given a query and a document... your mission is to perform the following steps.

1. Query Analysis: Think to reason and describe what information would...
2. Document Analysis: Discuss how the information provided by the document...
3. Relevance Annotation: ... annotate an integer score from 0 to 100. ...following guide:
   - 80-100 (Highly Relevant): ...
   - 60-80 (Relevant): ...
   - 40-60 (Moderately Relevant): ...
   - 20-40 (Slightly Relevant): ...
   - 0-20 (Irrelevant): ...

...conclude your entire response with the final relevance score...
<score>[...]</score>

Query:[Begin of Query]QUERY_INPUT[End of Query]
Document:[Begin of Document]DOCUMENT_INPUT[End of Document]
```

Figure 1: Relevance rubric for Retro*. The Relevance Placeholder allows users to specify the definition of relevance, while a 5-level criteria ensures consistent and interpretable scoring result.

## 2 RELATED WORK

**Reasoning Large Language Models**. Enhancing the reasoning capabilities of LLMs is crucial for tackling complex tasks. Techniques such as chain-of-thought prompting (Wei et al., 2022), which guide LLMs to reason step by step before producing the final answer, have shown significant performance gains. More advanced sampling strategies, including self-consistency (Wang et al., 2022), tree-of-thought (Yao et al., 2023), and Monte Carlo tree search (Xie et al., 2024), further enhance their reasoning quality and reliability. To unlock the full potential of LLM reasoning, recent research has increasingly explored reinforcement learning, which directly optimizes the model to generate high-quality reasoning trajectories (Jaech et al., 2024; Guo et al., 2025; Yang et al., 2025a).

**Reasoning-enhanced IR Methods**. Inspired by the recent advances in reasoning-capable LLMs, a growing body of research has begun adapting these paradigms to IR, with re-ranking tasks emerging as a primary focus. Preliminary studies have explored a variety of strategies, ranging from zero-shot prompting (Niu et al., 2024), to distilling reasoning trajectories from powerful reasoning LLMs through supervised fine-tuning (Weller et al., 2025; Yang et al., 2025b), and even reinforcement learning with carefully designed re-ranking rewards (Zhuang et al., 2025; Liu et al., 2025). Despite the promising progress of these approaches, there remains a lack of effective methods for directly estimating the strength of relevance between query and document. Moreover, existing methods face limitations in terms of test-time scalability and parallelism, which significantly hinder their overall accuracy and inference efficiency.

## 3 METHODOLOGY

In this section, we first introduce the basic framework of Retro*, including the rubric-based mechanism for relevance scoring and the score integration strategy for test-time scaling. We then describe how we optimize Retro*'s performance by reinforcement learning with tailored rewards.

### 3.1 RUBRIC-BASED RELEVANCE SCORING

Retro* is designed to both measure the absolute relevance of each document and re-rank a set of candidate documents. To this end, we introduce a rubric-based scoring mechanism that guides the model to perform reasoning on the relationship between a query and its candidate documents based on a well-defined relevance rubric, yielding relevance scores with interpretable meanings.

### 3.1.1 RELEVANCE RUBRIC

The relevance rubric ($\Gamma$) consists of two parts. One is the **Relevance Definition**, where the specific intent of the retrieval task is declared in the [Relevance Placeholder]. For example, we may specify an intent to retrieve proofs that are grounded in the same underlying theorem as the input. The other one is the **Scoring Criteria**, which defines the scope and rules for assigning relevance scores. In

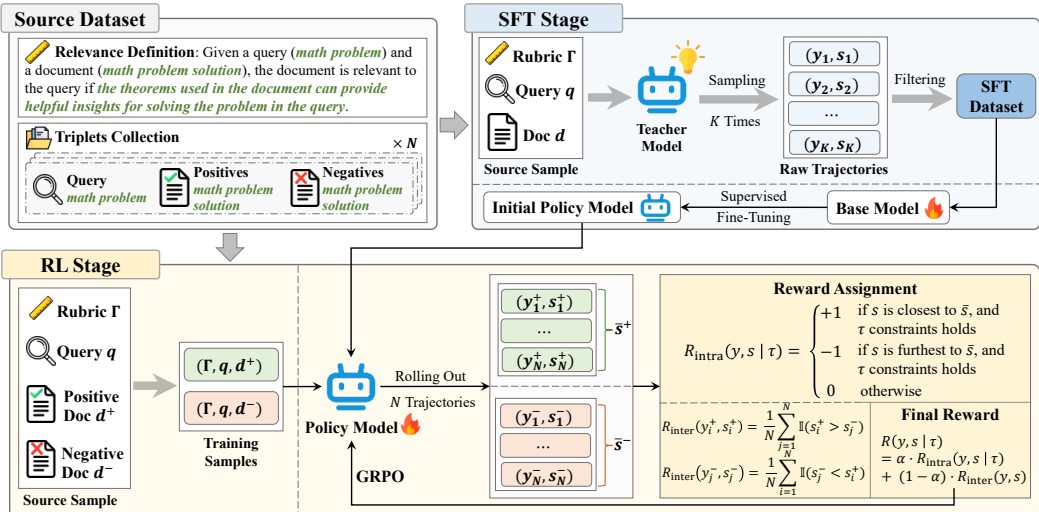

Figure 2: An overview of the two-stage training. **SFT**: the model is warmed up with a filtered data from a powerful teacher model. **RL**: the model is reinforced with the tailored composite reward.

our work, the relevance score is represented as an integer between 0 and 100, where higher scores indicate stronger relevance. The scope is partitioned into multiple intervals, each with explicit, interpretable meaning. For instance, a score within the range of 80-100 indicates that the candidate document is highly relevant and comprehensively addresses the information need of the query. An overview of the rubrics is shown in Figure 1, with the full rubrics provided in Appendix J.

Based on this well-defined rubric, Retro* is prompted to reason about the relevance between a given query ($q$) and a candidate document ($d$), generating a reasoning trajectory ($y$) along with a corresponding relevance score ($s$): $\text{Retro}(\Gamma, q, d) \to (y, s)$. See Appendix M for examples.

### 3.1.2 TEST-TIME SCALING VIA SCORE INTEGRATION

To enhance retrieval accuracy, Retro* leverages test-time scaling by sampling $K$ times for each query-document pair, resulting in a set of reasoning trajectories: $\{(y_1, s_1), \ldots, (y_K, s_K)\}$. A common approach to integrate these results is majority voting, where the most frequent score is chosen as the final score. While this approach works well for highly discrete outputs, it is not appropriate in Retro*, as drawing a reliable result would require a vast number of samples, making the process computationally expensive and impractical.

To this end, we propose a **Score Integration** strategy in a simple yet effective way: $\bar{s} \leftarrow \sum_K w_i * s_i / \sum_K w_i$, where $s_i$ is the relevance score from the i-th trajectory and $w_i$ is its associated weight. The weights can be set based on the generation likelihood of each trajectory. When likelihoods are unavailable or just for simplicity, a uniform weighting scheme, $w_i = 1/K$, can be used.

### 3.2 TRAINING STRATEGY

To optimize Retro*'s reasoning capabilities for reasoning-intensive document retrieval tasks, we propose a two-stage training strategy. The process starts with a supervised fine-tuning (SFT) stage to warm up the model, followed by a reinforcement learning (RL) stage for further performance enhancement. The overview of the training strategy is illustrated in Figure 2.

### 3.2.1 SUPERVISED FINE-TUNING

In the first stage, we perform SFT to equip the model with an initial reasoning ability and shape the model to generate concise and well-structured thoughts. The key of this stage lies in our approach to **Training Data Curation**, which involves two crucial steps:

**Data Sourcing**. We begin with a collection of query-document pairs $\{(q,d)_1, \ldots, (q,d)_M\}$. For each pair, we leverage a powerful teacher model $\mathbb{T}$ to reason about the relationship between $q$ and $d$ based on our relevance rubric: $\mathbb{T}(\Gamma, q, d) \rightarrow (y, s)$. Besides, the teacher model is also guided to keep its thoughts concise, with an explicit length control instruction.

**Data Filtering**. To obtain high-quality training data, we filter the raw trajectories with score integration. Specifically, we sample the teacher model $K$ times for each query-document pair, resulting in a group of raw trajectories $\{(y_1, s_1), \ldots, (y_K, s_K)\}$. We then integrate these scores to a score $\bar{s}$, as it serves as a reliable reference. Finally, the trajectory whose score is closest to $\bar{s}$ is selected, so that only one trajectory is used for each query-document pair, resulting in a curated dataset for SFT training: $\{(q, d, \hat{y}, \hat{s})_1, \ldots (q, d, \hat{y}, \hat{s})_M\}$, where $(\hat{y}, \hat{s})$ is the selected trajectory.

### 3.2.2 REINFORCEMENT LEARNING

In the second stage, we take the SFT model as the initial policy model and perform RL to further optimize its reasoning capability. As discussed earlier, Retro* is designed to achieve two key functionalities: accurately scoring the absolute relevance of individual documents and correctly ranking them by assigning higher scores to more relevant ones. To jointly optimize these functionalities, we introduce a composite reward that fully exploits the trajectories of each training sample during RL.

**Intra-Document Reward**. Intra-document reward is designed to improve the scoring accuracy for each individual document. It incentivizes the model to generate high-quality reasoning trajectories that produce stable and accurate relevance scores for the same query-document pair across multiple attempts. To this end, the reward is computed in two steps. 1) Rolling Out: For each query-document pair $(q, d)$, the model rolls out $N$ trajectories: $\{(y_1, s_1), \ldots, (y_N, s_N)\}$. 2) Reward Assignment: The integrated score $\bar{s}$ over all trajectories presents a more reliable result than each individual score in expectation. As such, the trajectories with scores closer to $\bar{s}$ are preferred, while those deviating further from $\bar{s}$ are disfavored. For simplicity and robustness, we adopt a ternary function, where the trajectory with the closest score is assigned a reward $+1$, the furthest one is assigned $-1$, while the others are assigned $0$. To prune the trivial cases where all trajectories already reach an agreement, we also introduce a threshold $\tau$ requiring a minimum gap between $\bar{s}$ and the furthest deviate score. The above reward is formulated as the following function:

$$R_{\text{intra}}(y, s) = \begin{cases} +1 & \text{if } s \text{ is closest to } \bar{s}, \text{ and the threshold } \tau \text{ constraints holds} \\ -1 & \text{if } s \text{ is furthest from } \bar{s}, \text{ and the threshold } \tau \text{ constraints holds} \\ 0 & \text{otherwise} \end{cases} . \quad (1)$$

**Inter-Document Reward**. Inter-document reward focuses on improving the reranking performance for a group of candidate documents. It incentivizes the model to assign higher relevance scores to more relevant documents, thereby resulting in a correct ranking for a given task. In our work, we sample one positive ($d^+$) and one negative ($d^-$) document from the candidate set for each query, and the reward is computed via the following steps: 1) Rolling Out: For each document $d \in \{d^+, d^-\}$, the model rolls out $N$ trajectories: $\{(y_1^d, s_1^d), \ldots, (y_N^d, s_N^d) \mid d \in \{d^+, d^-\}\}$. 2) Reward Assignment: The trajectories of both positive and negative samples are evaluated based on whether their scores correctly reflect the ranking. For a positive sample's trajectory $(y_i^+, s_i^+)$, the reward is the proportion of negative ones whose scores are dominated by $s_i^+$. Conversely, for a negative sample's trajectory $(y_j^-, s_j^-)$, the reward is the proportion of positive ones whose scores dominate $s_j^-$. These relationships are formally defined by the following equations:

$$R_{\text{inter}}(y_i^+, s_i^+) = \frac{1}{N} \sum_{j=1}^{N} \mathbb{I}(s_i^+ > s_j^-), \quad R_{\text{inter}}(y_j^-, s_j^-) = \frac{1}{N} \sum_{i=1}^{N} \mathbb{I}(s_j^- < s_i^+), \quad (2)$$

where $\mathbb{I}(\cdot)$ is the indicator function, and $N$ is the number of trajectories for each q-d sample.

The intra-document reward and inter-document reward are combined with a parameter $\alpha \in (0, 1)$ to obtain the composite reward: $R(y, s) = \alpha \cdot R_{\text{intra}}(y, s) + (1 - \alpha) \cdot R_{\text{inter}}(y, s)$, where $R_{\text{intra}}$ guides the model to assign reliable relevance scores for each individual document, and $R_{\text{inter}}$ incentivizes it to capture the relative rankings within a group of candidate documents. The composite reward is then optimized using the Group Relative Policy Optimization (GRPO) algorithm (Shao et al., 2024), with further analysis of the hyperparameter $\alpha$ provided in Appendix F.

## 4 EXPERIMENTS

In this section, we evaluate the overall performance of Retro* and the effectiveness of our detailed strategies. We focus on the following questions: 1) **RQ1**: How does Retro* perform in reasoning-intensive IR scenarios? 2) **RQ2**: How does Retro*'s performance compare to other non-reasoning and reasoning-enhanced re-ranking methods? 3) **RQ3**: How scalable is Retro* with the size of the backbone model and the number of test-time samples? 4) **RQ4**: How does Retro*'s parallelism capacity impact its efficiency compared to listwise and setwise methods, particularly as the number of candidate documents grows? 5) **RQ5**: Does Retro*'s rubric-based scoring mechanism provide a more reliable measurement of absolute relevance strength compared to those uncalibrated baselines? 6) **RQ6**: How do the key components of our proposed training strategy individually and collectively contribute to the effectiveness of Retro*?

### 4.1 EXPERIMENTAL SETUP

**Baselines and Backbone Models**. Our comparison includes several leading re-ranking models, categorized as non-reasoning and reasoning-enhanced approaches. For the non-reasoning baselines, we choose the pointwise model RankLLaMA (Ma et al., 2024) and the listwise model RankZephyr (Pradeep et al., 2023). In particular, RankLLaMA is built upon the Llama-2 (Touvron et al., 2023), and RankZephyr is built upon the Mistral-v0.1 (Jiang et al., 2023). For the reasoning-enhanced baselines, we compare against several powerful approaches built upon the Qwen2.5-Instruct family (Qwen et al., 2025). These include the fine-tuned models Rank1 (Weller et al., 2025), Rank-R1 (Zhuang et al., 2025), and ReasonRank (Liu et al., 2025), as well as the zero-shot method JudgeRank (Niu et al., 2024). The details for each model are provided in Appendix B. To ensure a fair comparison, we align with the baselines by employing Qwen2.5-7B-Instruct and Qwen2.5-32B-Instruct as our backbone models. In addition, we also provide results trained on other backbone models in Appendix G.

**Evaluation Settings**. We evaluate all baselines and Retro* on the BRIGHT (Su et al., 2024) benchmark, which is designed for reasoning-intensive IR. We define the relevance following the BRIGHT benchmark and provide the detailed definitions in Appendix L. Our Retro* evaluation is conducted using the FlagEmbedding (Xiao et al., 2024) framework, and inference is accelerated with SGLang (Zheng et al., 2024a). For simplicity, we employ a uniform weighting scheme for our score integration strategy, which we denote as mean-score@k. To provide a more challenging and comprehensive evaluation, we use the powerful BGE-Reasoner-Embed-0821[1] (BGE-Reasoner-Embed) as the first-stage retriever, which is the current state-of-the-art embedding model on the BRIGHT benchmark. We utilize its publicly available search results, as they provide more positive document candidates. In our evaluation, all methods re-rank the top-100 documents retrieved by this embedder with the original query, and we report nDCG@10 as the performance metric.

**Training Data**. We construct our training data based on the BGE-Reasoner-Data, originally released with the BGE-Reasoner-Embed model. BGE-Reasoner-Data is a synthetic, reasoning-centric retrieval dataset that covers all domains in the BRIGHT benchmark. Specifically, its queries are constructed using a document-to-query generation pipeline, where documents from each BRIGHT domain are used as source material to generate synthetic reasoning-intensive queries. These queries are then used to retrieve the candidate documents, which are finally annotated as positives or negatives using powerful LLMs. As a result, the dataset provides high-quality synthetic queries and relevance annotations, suitable for training reasoning-intensive retrieval models. From this dataset, we sample 500 queries per dataset for SFT and 1,000 queries per dataset for RL. Each query is associated with one positive and one negative document, yielding a total of 12,000 training samples for SFT and 24,000 for RL. For SFT, we obtain the reasoning trajectories from a strong teacher model, Qwen3-235B-A22B (Yang et al., 2025a). For each training sample, we sample eight trajectories from the teacher model and train the model with only one selected trajectory. Also, we augment the original prompt template (shown in Figure 8) with an explicit instruction asking the teacher model to keep its reasoning concise within 512 tokens as described in Section 3.2.1. Generating all trajectories for SFT under this setting on 8 NVIDIA H100 GPUs takes less than 6 hours. We discuss

---

[1]BGE-Reasoner-Embed-0821: `https://github.com/FlagOpen/FlagEmbedding/tree/master/research/BGE_Reasoner`

Table 1: Performance on BRIGHT benchmark (nDCG@10), where all methods re-rank the top-100 documents retrieved by BGE-Reasoner-Embed. Retro* achieves the leading performance, with test-time scaling further pushing its edge.

| Models | Methods | Avg. | StackExchange | | | | | | | Coding | | Theorem-based | | |
| --- | --- | --- | --- | --- | --- | --- | --- | --- | --- | --- | --- | --- | --- | --- |
| | | | Bio. | Earth. | Econ. | Psy. | Rob. | Stack. | Sus. | Leet. | Pony | AoPS | TheoQ. | TheoT. |
| BGE-Reasoner-Embed | Retriever | 32.5 | 42.6 | 42.6 | 27.8 | 37.3 | 26.4 | 29.6 | 30.6 | 36.9 | 25.7 | 9.8 | 34.9 | 46.1 |
| *Non-Reasoning Re-Ranking Baselines* | | | | | | | | | | | | | | |
| RankLLaMA (7B) | Pointwise | 19.7 | 20.8 | 31.7 | 11.1 | 19.1 | 18.2 | 10.9 | 20.0 | 16.7 | 57.9 | 6.3 | 10.3 | 13.4 |
| RankLLaMA (14B) | Pointwise | 21.3 | 26.4 | 34.1 | 19.1 | 24.4 | 19.9 | 15.6 | 22.0 | 17.2 | 41.8 | 6.9 | 11.8 | 16.6 |
| RankZephyr (7B) | Listwise | 20.8 | 27.1 | 22.7 | 19.9 | 21.9 | 13.2 | 8.6 | 22.3 | 22.1 | 50.6 | 7.8 | 21.3 | 12.4 |
| *Reasoning-Enhanced Re-Ranking Baselines* | | | | | | | | | | | | | | |
| JudgeRank (7B) | Pointwise | 12.7 | 14.7 | 16.9 | 10.8 | 9.5 | 7.7 | 7.1 | 10.8 | 11.4 | 16.7 | 5.8 | 9.2 | 31.9 |
| JudgeRank (32B) | Pointwise | 17.4 | 21.4 | 26.0 | 13.1 | 12.9 | 10.5 | 15.8 | 16.2 | 19.7 | 19.9 | 5.2 | 16.4 | 31.5 |
| Rank1 (7B) | Pointwise | 25.9 | 43.2 | 33.1 | 21.6 | 33.2 | 19.8 | 20.7 | 28.8 | 7.5 | 28.2 | 8.3 | 30.9 | 35.7 |
| Rank1 (32B) | Pointwise | 29.7 | 46.0 | 31.5 | 27.2 | 33.9 | 19.5 | 22.8 | 33.1 | 17.2 | 36.2 | 15.8 | 28.2 | 44.6 |
| Rank-R1 (7B) | Setwise | 25.8 | 34.9 | 29.7 | 26.8 | 35.5 | 21.6 | 17.5 | 29.4 | 24.2 | 15.6 | 5.8 | 25.1 | 43.1 |
| Rank-R1 (14B) | Setwise | 31.8 | 46.6 | 43.3 | 28.9 | 42.2 | 30.3 | 26.5 | 38.0 | 18.5 | 18.5 | 10.9 | 33.9 | 44.0 |
| ReasonRank (7B) | Listwise | 33.5 | 48.9 | 44.4 | 33.1 | 44.5 | 31.1 | 30.8 | 39.1 | 22.8 | 21.8 | 7.7 | 37.1 | 40.1 |
| ReasonRank (32B) | Listwise | 36.6 | 54.9 | 50.1 | 38.0 | 49.1 | 35.1 | 35.3 | 44.6 | 23.2 | 15.8 | 10.1 | 39.2 | 44.4 |
| *Our Models* | | | | | | | | | | | | | | |
| Retro* (7B) | Pointwise | 36.6 | 53.7 | 55.9 | 35.6 | 47.9 | 34.0 | 35.6 | 39.3 | 17.6 | 29.8 | 9.6 | 35.4 | 45.0 |
| Retro* (32B) | Pointwise | 38.5 | 59.4 | 58.3 | 41.8 | 48.3 | 37.0 | 38.4 | 44.1 | 13.9 | 27.8 | 10.3 | 36.6 | 45.7 |
| *Test-Time Scaling (Mean-Score@16)* | | | | | | | | | | | | | | |
| Retro* (7B) | Pointwise | 38.7 | 58.4 | 59.2 | 35.0 | 49.3 | 33.9 | 37.7 | 41.1 | 18.8 | 33.5 | 10.7 | 40.2 | 46.7 |
| Retro* (32B) | Pointwise | 40.6 | 61.4 | 61.3 | 41.7 | 50.8 | 37.8 | 41.9 | 46.8 | 13.5 | 32.0 | 12.2 | 39.7 | 47.8 |

Table 2: Performance (nDCG@10) on the BRIGHT benchmark using different first-stage retrievers: BM25, ReasonIR, where the top-100 documents are re-ranked by Retro*

| Models | BM25 | Retro* (7B) | | Retro* (32B) | | ReasonIR | Retro* (7B) | | Retro* (32B) | |
| --- | --- | --- | --- | --- | --- | --- | --- | --- | --- | --- |
| | | @1 | @16 | @1 | @16 | | @1 | @16 | @1 | @16 |
| Avg. | 27.0 | 35.3 | 37.0 | 36.6 | 38.5 | 30.6 | 36.8 | 38.4 | 37.4 | 39.5 |

the impact of this length control instruction in Appendix H. The training details of SFT and RL are available in Appendix C.

## 4.2 MAIN RESULTS (FOR RQ1 AND RQ2)

Results in Table 1 clearly demonstrate that our Retro* models achieve state-of-the-art performance on the BRIGHT benchmark. Specifically, our Retro* (7B) achieves an average nDCG@10 of 36.6, substantially outperforming all other 7B-scale non-reasoning and reasoning-enhanced baselines, and surpassing ReasonRank (7B) by 3.1 points. Scaling up to the 32B extends this lead, achieving a remarkable score of 38.5, and surpassing ReasonRank (32B) by 1.9 points.

Furthermore, with test-time scaling, the performance of our models is further improved. Specifically, averaging over 16 sampling scores (mean-score@16) boosts the 7B model to 38.7, while enhancing the 32B model to 40.6. Notably, the scaled 7B model (38.7) not only surpasses all baselines but also exceeds the performance of our own standard 32B model (38.5), powerfully demonstrating both the effectiveness and potential of test-time scaling.

Additionally, we evaluate its performance on candidates from different first-stage retrievers, namely BM25 and ReasonIR (Shao et al., 2025). As shown in Table 2, our model's performance remains consistent across different first-stage retrievers, which demonstrates the effectiveness and robustness of its ranking capabilities. More detailed results for BM25 and ReasonIR are provided in Appendix D. To further assess the out-of-domain generalizability of our rubric, we evaluate Retro* on R2MED (Li et al., 2025), a biomedical reasoning retrieval benchmark that lies outside the scientific, programming, and mathematical domains used during training. We provide detailed analysis of these results in Appendix I. In addition to Retro*'s effectiveness in reasoning-intensive scenarios, we also empirically validate its generalizability on traditional IR scenarios using the BEIR (Thakur et al., 2021), with detailed results provided in Appendix E.

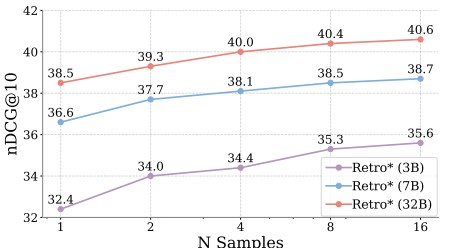 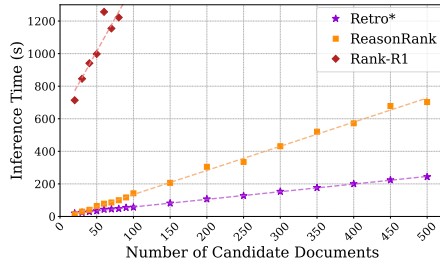

Figure 3: **(Left):** average performance (nDCG@10) on BRIGHT benchmark. Retro*'s re-ranking performance consistently improves with increased model scale and test-time samples. **(Right):** inference time on the TheoT. dataset from the BRIGHT benchmark. Retro* exhibits a significantly lower time latency than other methods as the number of candidate documents increases.

## 4.3 Performance of Scaling (For RQ3)

To investigate the scalability of Retro*, we conduct experiments under two scaling settings: scaling the model size and scaling the number of sampling trajectories at test-time.

**Model Size Scaling**. As shown in Figure 3, the performance of Retro* improves consistently as the model size increases. Specifically, under single-sample inference (N = 1), the nDCG@10 rises substantially from 32.4 for the 3B model to 36.6 for the 7B, and further to 38.5 for the 32B. This scaling trend remains consistent across different numbers of sampling generations. These findings highlight the scaling benefits of larger LLMs, indicating that larger and more capable LLMs exhibit stronger reasoning capabilities, enabling better comprehension of queries, more nuanced document analysis, and more accurate relevance estimates within our framework.

**Test-Time Scaling**. As shown in Figure 3, the performance of Retro* improves consistently as the number of sampling trajectories increases. Specifically, for the 7B model, nDCG@10 improves from 36.6 (N = 1) to 38.7 (N = 16), with consistent gains at intermediate sampling steps. This scaling trend remains consistent across all model sizes. These results underscore the effectiveness of test-time scaling, demonstrating that integrating multiple reasoning trajectories leads to more accurate and reliable relevance scoring.

## 4.4 Efficiency of Parallelism (for RQ4)

Listwise and setwise re-ranking methods have limited parallelism capabilities, leading to significant latency when processing a large group of candidate documents. This inefficiency stems directly from their inherently sequential processing nature, which lacks the capability for effective document-level batching. In contrast, the pointwise approach evaluates each query-document pair independently. This property enables massive parallelism and is inherently more efficient in terms of both inference time and throughput. To assess the practical efficiency gains from parallel processing, we compare the inference times of models with different re-ranking manners across varying numbers of candidate documents. This experiment is conducted on the TheoT. dataset using 4 NVIDIA H100 GPUs.

The results presented in Figure 3 empirically confirm the benefits of this massive parallelism. Retro* demonstrates a significantly slower increase in inference time as the number of candidate documents grows. In contrast, the setwise model Rank-R1 exhibits extremely poor scalability, with inference time becoming prohibitively expensive even for a modest number of documents. Although the listwise model ReasonRank supports query-level batching, it still experiences a much steeper increase in latency. These findings demonstrate that the parallelism capability of Retro* delivers significant efficiency, making it a far more practical and scalable solution for real-world IR applications.

## 4.5 Scoring of Relevance (for RQ5)

A primary limitation of existing models is their inability to provide a direct and interpretable measurement of relevance. Although pointwise models like RankLLaMA and Rank1 can produce absolute scores, their logit-based or probability-based scores lack clear and interpretable functionality

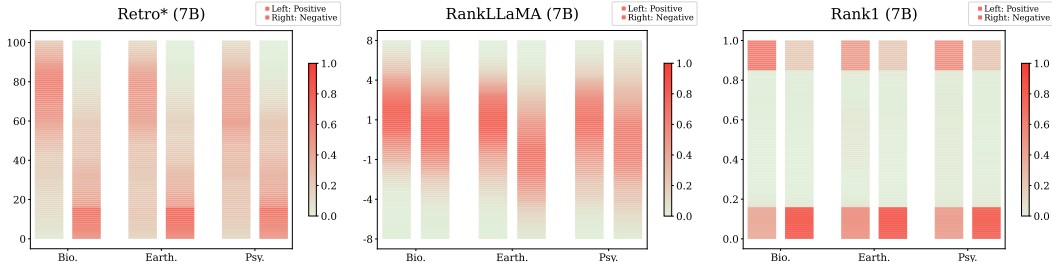

Figure 4: Score distributions from pointwise models on a sample of positive and negative documents on the BRIGHT benchmark. The intensity of the color represents the density of scores, with darker hues indicating a larger proportion of documents are assigned scores in that range.

Table 3: Ablation study on the key components of our training strategy.

| Models | Avg. | StackExchange | | | | | | | Coding | | Theorem-based | | |
| --- | --- | --- | --- | --- | --- | --- | --- | --- | --- | --- | --- | --- | --- |
| | | Bio. | Earth. | Econ. | Psy. | Rob. | Stack. | Sus. | Leet. | Pony | AoPS | TheoQ. | TheoT. |
| Qwen2.5-7B-Instruct | 22.9 | 39.9 | 41.2 | 21.0 | 31.4 | 17.0 | 16.9 | 22.7 | 12.1 | 15.7 | 3.9 | 14.2 | 38.7 |
| + SFT + RL (Composite Reward) | 36.6 | 53.7 | 55.9 | 35.6 | 47.9 | 34.0 | 35.6 | 39.3 | 17.6 | 29.8 | 9.6 | 35.4 | 45.0 |
| + only-SFT | 30.1 | 46.9 | 51.3 | 29.1 | 37.4 | 24.4 | 28.4 | 35.0 | 15.7 | 20.6 | 7.9 | 27.9 | 36.6 |
| + only-RL (Composite Reward) | 35.1 | 56.7 | 52.9 | 33.2 | 45.1 | 28.1 | 32.5 | 35.8 | 21.2 | 36.3 | 8.6 | 29.8 | 41.0 |
| + SFT + RL (Intra-Reward) | 33.2 | 49.4 | 51.8 | 29.9 | 44.2 | 27.6 | 33.4 | 36.4 | 20.1 | 23.3 | 8.6 | 32.0 | 41.3 |
| + SFT + RL (Inter-Reward) | 30.8 | 43.6 | 48.7 | 30.8 | 36.7 | 25.6 | 28.5 | 37.7 | 13.1 | 28.6 | 5.6 | 30.6 | 40.6 |

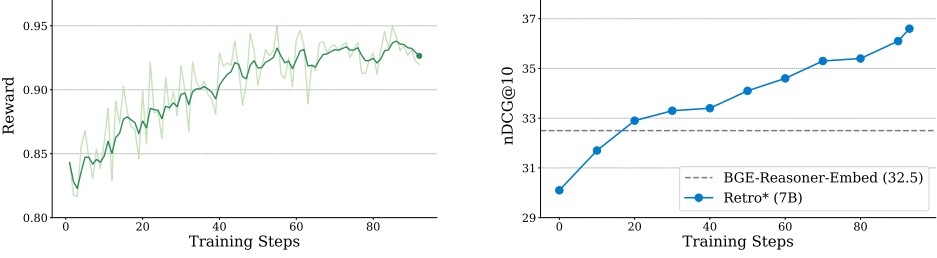

Figure 5: Training dynamics of Retro* (7B) during the RL stage. **(Left):** The training reward steadily increases over training steps. **(Right):** along with the improved reward, the retrieval accuracy (nDCG@10) improves consistently with the training steps.

for measuring relevance. As shown in Figure 4, this deficiency is evident. The score distributions of RankLLaMA for positive and negative documents are heavily mixed, making it impossible to establish a meaningful threshold to filter out the negative documents. Similarly, while Rank1's scores exhibit a binary tendency, a significant number of negative documents still receive high probability scores, undermining the reliability of its relevance scoring.

In contrast, Retro* demonstrates a clean and consistent separation between positive and negative documents. Aligned with its explicit relevance rubric, positive documents consistently receive high scores (typically above 60), while negative documents are assigned low scores. This clear separation demonstrates that Retro* not only excels at relative ranking but also successfully provides the crucial functionality of relevance measurement, a key advantage for downstream applications.

## 4.6 ABLATION STUDY (FOR RQ6)

We conduct ablation studies on the 7B model to validate the contribution of each component in our training strategy. All variants follow the same training data and process as the main experiments, differing only in the component being ablated. Results are shown in the Table 3. As a baseline, the backbone Qwen2.5-7B-Instruct achieves an average nDCG@10 of 22.9 without any training. After our two-stage training with the composite reward, the performance significantly improves to 36.6.

The following experiments analyze the contribution of each component to this overall improvement of 13.7 points.

**Effect of the Two-Stage Training**. Training with SFT alone improves performance to 30.1 (+7.2), indicating that learning from the teacher model's reasoning trajectories provides the model with an initial ranking ability. In comparison, training with RL alone achieves a higher score of 35.1 (+12.2), also outperforming other baselines of similar model size, demonstrating the importance of ranking-oriented rewards in enhancing the model's performance. Furthermore, training with both SFT and RL produces a better result of 36.6 (+13.7). This result indicates that combining the two stages helps the model perform more effective exploration and exploitation during the RL stage, while also demonstrating that the teacher-generated SFT trajectories are not strictly required, as smaller models trained with RL alone can already achieve competitive performance.

**Effect of the Composite Rewards**. We analyze the contribution of both intra-reward and inter-reward under the full two-stage training setup. Training with intra-reward improves performance from 30.1 to 33.2, while using only the inter-reward has a limited improvement to 30.8. In contrast, combining both rewards results in a final score of 36.6. These results demonstrate that without intra-reward, the model cannot generate a reliable trajectory for a given query-document, which introduces noisy learning signals and hinders effective learning. Without inter-reward, the model struggles to distinguish between the positive and negative documents. With composite reward, the model learns more steadily, as illustrated in Figure 5, which shows that both the training reward and nDCG@10 on the BRIGHT benchmark steadily increase throughout the training process.

## 5 CONCLUSION

This paper proposes Retro*, a novel approach for reasoning-intensive document retrieval. Retro* introduces a rubric-based relevance scoring mechanism, which enables the LLMs to reason about the relationship between a task and a document based on well-defined criteria. Additionally, Retro* also supports test- time scaling by combining multiple reasoning trajectories via score integration, which produces more reliable relevance estimates. To optimize Retro*'s reasoning capabilities for document retrieval tasks, we propose a two-stage training strategy which includes SFT for warm up and RL with composite rewards for scoring and ranking functionalities. Empirical experiments on the BRIGHT benchmark demonstrate that Retro* outperforms existing methods and achieves state-of-the-art performance, with flexible test-time scalability and massive parallelism capability.

### AUTHOR CONTRIBUTIONS

As the core contributors, Junwei Lan and Jianlyu Chen started this work collaboratively. Junwei was responsible for the model architecture, training algorithm, training and evaluation, experiments, and manuscript writing. Jianlyu contributed to the refinement and optimization of the model architecture, the model evaluation framework, and part of the manuscript writing.

### ACKNOWLEDGMENTS

This work is supported by grants from Scientific Research Innovation Capability Support Project for Young Faculty and the National Natural Science Foundation of China (No. U24A20253).

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

OVERVIEW OF APPENDIX

- Appendix A: The Use of LLMs
- Appendix B: Baselines
- Appendix C: Training Details
- Appendix D: Detailed BM25 and ReasonIR Results on BRIGHT
- Appendix E: Evaluation on BEIR
- Appendix F: Analysis of the Hyperparameter $\alpha$
- Appendix G: Different Backbone Models
- Appendix H: Analysis of the Length Control Instruction for Distillation
- Appendix I: Out-of-Domain Generalization on R2MED
- Appendix J: Rubrics of Rubric-based Relevance Scoring
- Appendix K: Analysis of the Necessity of Reasoning
- Appendix L: Task Relevance Definitions for Different Benchmarks
- Appendix M: Reasoning Examples

## A    THE USE OF LLMS

We used the LLMs to assist with writing. Specifically, their use included grammar checking, rephrasing for clarity, and textual polishing. Additionally, we utilized the LLMs to help draft Python codes for plotting figures. The LLMs used for these purposes include Gemini 2.5 Pro[2], ChatGPT[3], DeepSeek-V3.1[4].

## B    BASELINES

This section provides additional details for each of the baseline models used in our comparative analysis. As outlined in the experimental setup 4.1, we categorize these models into two groups: non-reasoning and reasoning-enhanced approaches. For each baseline listed below, we describe its core methodology and implementation details:

- **RankLLaMA**: A pointwise re-ranking model that jointly encodes the query and document, generating a scalar relevance score by projecting the representation of the end-of-sequence token through a linear layer. The model is trained with a contrastive loss.

- **RankZephyr**: A listwise re-ranking model that takes a query and a set of candidate documents as input and outputs their relative ranking order. It employs a sliding window strategy to to handle large lists of candidates. The model is trained on MS MARCO (Bajaj et al., 2016) dataset via knowledge distillation from GPT-3.5 (Ouyang et al., 2022) and GPT-4 (Achiam et al., 2023).

- **JudgeRank**: A zero-shot, reasoning-enhanced pointwise re-ranking method. It utilizes carefully designed prompts to guide LLMs through explicit reasoning steps before producing the final relevance judgment ("yes" or "no").

- **Rank1**: A reasoning-enhanced pointwise re-ranking model. For each query-document pair, the model generates a reasoning process and outputs a binary relevance judgment ("yes" or "no"). Rank1 is trained on reasoning trajectories generated by DeepSeek-R1 (Guo et al., 2025) on the MS MARCO dataset.

- **Rank-R1**: A reasoning-enhanced setwise (Zhuang et al., 2024) re-ranking model. Given a query and a candidate set, the model selects the most relevant document and applies a heap-sort procedure to obtain top-k results. The model is trained on the MS MARCO dataset with the GRPO algorithm.

---

[2]https://gemini.google.com
[3]https://chatgpt.com
[4]https://chat.deepseek.com

Table 4: Hyperparameters for the SFT stage.

| Learning Rate | Epochs | WarmUp Ratio | LR Scheduler | Batch Size Per-Device | Gradient Accumulation |
|---|---|---|---|---|---|
| $10^{-5}$ | 1 | 0.1 | cosine | 1 | 8 |

Table 5: Hyperparameters for the RL stage with GRPO.

| $\alpha$ | $\tau$ | N | Learning Rate | Epochs | Batch Size | Mini Batch Size | Micro Batch Size Per-Device | KL Coefficient | Temperature |
|---|---|---|---|---|---|---|---|---|---|
| 0.75 | 20 | 8 | $10^{-6}$ | 1 | 256 | 256 | 8 | 0.005 | 1.0 |

Table 6: Detailed nDCG@10 results for Retro* on the BRIGHT benchmark. Retro* re-ranks the top-100 documents retrieved by either BM25 or ReasonIR. The retrieval stage utilized a GPT-4 reasoning query, while the re-ranking stage utilized the original query.

| Models | Avg. | StackExchange | | | | | | | Coding | | Theorem-based | | |
|---|---|---|---|---|---|---|---|---|---|---|---|---|---|
| | | Bio. | Earth. | Econ. | Psy. | Rob. | Stack. | Sus. | Leet. | Pony | AoPS | TheoQ. | TheoT. |
| BM25 | 27.0 | 53.6 | 54.1 | 24.3 | 38.7 | 18.9 | 27.7 | 26.3 | 19.3 | 17.6 | 3.9 | 19.2 | 20.8 |
| Retro* (7B) | 35.3 | 56.4 | 56.1 | 30.8 | 46.4 | 31.6 | 36.7 | 39.8 | 22.7 | 28.2 | 7.3 | 27.6 | 40.4 |
| Retro* (32B) | 36.6 | 61.6 | 59.4 | 35.9 | 47.8 | 33.6 | 35.6 | 43.2 | 18.8 | 29.2 | 6.6 | 28.1 | 39.3 |
| *Test-Time Scaling (Mean-Score@16)* | | | | | | | | | | | | | |
| Retro* (7B) | 37.0 | 60.7 | 59.0 | 31.9 | 48.2 | 32.1 | 34.9 | 40.8 | 25.2 | 33.2 | 8.5 | 29.8 | 40.1 |
| Retro* (32B) | 38.5 | 64.0 | 61.6 | 36.9 | 50.1 | 33.9 | 37.7 | 45.2 | 19.9 | 32.3 | 8.5 | 30.6 | 41.5 |
| ReasonIR | 30.6 | 43.4 | 43.0 | 33.1 | 39.6 | 20.9 | 31.0 | 27.0 | 31.6 | 19.5 | 7.4 | 33.9 | 36.9 |
| Retro* (7B) | 36.8 | 54.7 | 54.5 | 33.5 | 47.7 | 32.5 | 40.1 | 41.2 | 22.7 | 28.9 | 8.3 | 34.1 | 43.7 |
| Retro* (32B) | 37.4 | 60.7 | 56.1 | 37.6 | 47.7 | 31.3 | 38.6 | 45.7 | 19.0 | 27.3 | 9.0 | 33.0 | 43.2 |
| *Test-Time Scaling (Mean-Score@16)* | | | | | | | | | | | | | |
| Retro* (7B) | 38.4 | 60.4 | 57.4 | 34.8 | 48.2 | 31.6 | 37.9 | 41.9 | 24.9 | 32.1 | 9.0 | 38.1 | 44.0 |
| Retro* (32B) | 39.5 | 64.7 | 57.5 | 38.5 | 49.7 | 31.7 | 42.6 | 47.3 | 19.4 | 30.2 | 10.0 | 37.8 | 44.8 |

- **ReasonRank**: A reasoning-enhanced listwise re-ranking model. It introduces an automated data synthesis framework to generate high-quality reasoning-intensive training data. The model is trained on this synthesized dataset in two stages: SFT with reasoning trajectories distilled from DeepSeek-R1, followed by GRPO with a multi-view re-ranking reward.

## C  TRAINING DETIALS

For the SFT stage, we use the LLaMA-Factory framework (Zheng et al., 2024b), fine-tuning both the 7B and 32B models on a single node equipped with 8 NVIDIA H100 GPUs. For the RL stage, we employ the GRPO algorithm through the VeRL framework (Sheng et al., 2025). In our main experiments, we set the hyperparameter $\alpha$ to 0.75 and $\tau$ to 20, as 20 represents a meaningful interval between relevance levels in our rubrics 3.1.1. An ablation study on the impact of different $\alpha$ values is presented in Appendix F. The 7B model is trained on a single node, while the 32B model is trained across two nodes. Detailed hyperparameters for the SFT and RL stages are provided in Table 4 and Table 5, respectively.

## D  DETAILED BM25 AND REASONIR RESULTS ON BRIGHT

This section provides the complete, per-dataset results on the BRIGHT benchmark, corresponding to the summarized analysis in the main body. Table 6 details the nDCG@10 scores of Retro* when re-ranking the top-100 documents retrieved by two different first-stage retrievers: BM25 and ReasonIR. The results highlight the model's robust and consistent performance across various domains, regardless of the initial retrieval method.

ReasonIR further introduces a zero-shot re-ranking method (QwenRerank) based on Qwen2.5-Instruct-32B, and we conduct a detailed comparison between this method and Retro*. However, QwenRerank computes its final ranking scores by fusing normalized retriever and re-ranker outputs with $0.5 \times s_{\text{reranker}} + 0.5 \times s_{\text{retriever}}$, meaning that its reported performance reflects the joint effect of

Table 7: Comparison of Retro* with QwenRerank from ReasonIR under two evaluation settings, where all methods re-rank the top-100 documents retrieved by ReasonIR. Retro* consistently achieves higher performance in both settings. Results marked with [†] are reported by ReasonIR.

| Models | Avg. | StackExchange | | | | | | | Coding | | Theorem-based | | |
|---|---|---|---|---|---|---|---|---|---|---|---|---|---|
| | | Bio. | Earth. | Econ. | Psy. | Rob. | Stack. | Sus. | Leet. | Pony | AoPS | TheoQ. | TheoT. |
| ReasonIR | 30.6 | 43.4 | 43.0 | 33.1 | 39.6 | 20.9 | 31.0 | 27.0 | 31.6 | 19.5 | 7.4 | 33.9 | 36.9 |
| *Re-ranking w/o Score Fusion* | | | | | | | | | | | | | |
| QwenRerank[†] (32B) | 30.3 | 50.2 | 47.5 | 23.6 | 35.8 | 24.7 | 28.1 | 29.9 | 30.3 | 26.5 | 5.5 | 20.7 | 40.8 |
| Retro* (7B) | 36.8 | 54.7 | 54.5 | 33.5 | 47.7 | 32.5 | 40.1 | 41.2 | 22.7 | 28.9 | 8.3 | 34.1 | 43.7 |
| Retro* (32B) | 37.4 | 60.7 | 56.1 | 37.6 | 47.7 | 31.3 | 38.6 | 45.7 | 19.0 | 27.3 | 9.0 | 33.0 | 43.2 |
| Retro* (7B, Mean-Score@16) | 38.4 | 60.4 | 57.4 | 34.8 | 48.2 | 31.6 | 37.9 | 41.9 | 24.9 | 32.1 | 9.0 | 38.1 | 44.0 |
| Retro* (32B, Mean-Score@16) | 39.5 | 64.7 | 57.5 | 38.5 | 49.7 | 31.7 | 42.6 | 47.3 | 19.4 | 30.2 | 10.0 | 37.8 | 44.8 |
| *Re-ranking w/ Score Fusion* | | | | | | | | | | | | | |
| QwenRerank[†] (32B) | 36.9 | 58.2 | 53.2 | 32.0 | 43.6 | 28.8 | 37.6 | 36.0 | 33.2 | 34.8 | 7.9 | 32.6 | 45.0 |
| Retro* (7B) | 38.4 | 54.2 | 55.9 | 35.0 | 49.7 | 32.4 | 40.2 | 41.7 | 26.1 | 32.7 | 10.9 | 36.9 | 45.0 |
| Retro* (32B) | 39.6 | 60.0 | 58.2 | 38.4 | 51.1 | 31.4 | 39.2 | 45.3 | 25.0 | 33.4 | 10.5 | 37.0 | 45.4 |
| Retro* (7B, Mean-Score@16) | 39.0 | 56.8 | 57.1 | 36.0 | 49.8 | 31.7 | 38.2 | 42.3 | 27.1 | 34.1 | 10.4 | 39.2 | 44.9 |
| Retro* (32B, Mean-Score@16) | 40.2 | 60.1 | 58.4 | 38.5 | 51.4 | 32.1 | 41.5 | 46.3 | 25.2 | 33.3 | 11.1 | 38.4 | 46.2 |

Table 8: nDCG@10 results on the BEIR benchmark, where all methods re-rank the top-100 documents retrieved by BM25 provided by Lin et al. (2021). The retrieval stage and the re-ranking stage both utilize the original query. Results marked with [†] are reported by ReasonRank.

| Models | Methods | Avg. | TREC-COVID | DBPedia | SciFact | NFCorpus | Signal-1M | Robust04 | TREC-NEWS |
|---|---|---|---|---|---|---|---|---|---|
| BM25 | Retriever | 43.7 | 59.5 | 31.8 | 67.9 | 33.8 | 33.0 | 40.7 | 39.5 |
| *Non-Reasoning Re-Ranking Baselines* | | | | | | | | | |
| RankZephyr[†] (7B) | Listwise | 54.1 | 82.9 | 44.4 | 75.4 | 38.3 | 31.4 | 53.7 | 52.8 |
| *Reasoning-Enhanced Re-Ranking Baselines* | | | | | | | | | |
| Rank1[†] (7B) | Pointwise | 50.7 | 79.0 | 35.8 | 73.3 | 37.5 | 25.4 | 57.1 | 47.7 |
| Rank1[†] (32B) | Pointwise | 51.0 | 80.6 | 34.8 | 74.8 | 37.3 | 25.6 | 58.3 | 45.6 |
| Rank-R1[†] (7B) | Setwise | 53.6 | 83.7 | 42.3 | 72.2 | 38.9 | 33.1 | 54.5 | 50.6 |
| Rank-R1[†] (14B) | Setwise | 54.6 | 84.6 | 44.1 | 76.0 | 38.6 | 33.0 | 56.9 | 49.2 |
| ReasonRank[†] (7B) | Listwise | 54.4 | 82.0 | 46.0 | 75.6 | 39.6 | 31.4 | 55.4 | 50.5 |
| ReasonRank[†] (32B) | Listwise | 55.4 | 83.2 | 45.7 | 77.2 | 40.0 | 31.1 | 58.7 | 52.2 |
| Retro* (7B) | Pointwise | 55.8 | 84.7 | 45.9 | 77.1 | 37.1 | 31.2 | 64.7 | 49.7 |
| *Test-Time Scaling (Mean-Score@16)* | | | | | | | | | |
| Retro* (7B) | Pointwise | 56.8 | 85.4 | 46.7 | 78.5 | 38.1 | 32.1 | 65.8 | 51.0 |

both retriever and re-ranker rather than its standalone re-ranking capability. To ensure a fair comparison, we compare both models under the same evaluation settings. As shown in Table 7, Retro* consistently outperforms QwenRerank across all evaluation settings, indicating substantially stronger re-ranking ability. These results suggest that most of the gains attributed to QwenRerank largely come from the strong retriever ReasonIR, whereas Retro* achieves higher performance purely from its re-ranking capability.

# E    EVALUATION ON BEIR

To assess the generalizability of our model, we further evaluate its performance in traditional retrieval scenarios. Following ReasonRank (Liu et al., 2025), we select seven datasets from the BEIR benchmark (Thakur et al., 2021), for their relatively small number of queries. In the retrieval stage, BM25 is applied with the original query to obtain candidate documents, after which all re-ranking methods re-rank the top-100 BM25 results. We report nDCG@10 as the performance metric.

In this scenario, we utilize the MS MARCO dataset as our training data. Specifically, we construct 24,000 query-document pairs for both SFT and RL training. The RL stage is trained for 3 epochs, while the other training settings are consistent with Section 4.1.

The evaluation results on BEIR, as shown in Table 8, demonstrate the strong generalizability of Retro* to traditional re-ranking tasks. The Retro* (7B) model achieves an average nDCG@10 of 55.8, outperforming all non-reasoning and reasoning-enhanced baselines. These results confirm

Table 9: nDCG@10 results on the BRIGHT benchmark with models trained using varying $\alpha$ values.

| Models | Avg. | StackExchange | | | | | | | Coding | | Theorem-based | | |
|---|---|---|---|---|---|---|---|---|---|---|---|---|---|
| | | Bio. | Earth. | Econ. | Psy. | Rob. | Stack. | Sus. | Leet. | Pony | AoPS | TheoQ. | TheoT. |
| + only-SFT | 30.1 | 46.9 | 51.3 | 29.1 | 37.4 | 24.4 | 28.4 | 35.0 | 15.7 | 20.6 | 7.9 | 27.9 | 36.6 |
| + SFT + RL ($\alpha = 0.00$) | 30.8 | 43.6 | 48.7 | 30.8 | 36.7 | 25.6 | 28.5 | 37.7 | 13.1 | 28.6 | 5.6 | 30.6 | 40.6 |
| + SFT + RL ($\alpha = 1.00$) | 33.2 | 49.4 | 51.8 | 29.9 | 44.2 | 27.6 | 33.4 | 36.4 | 20.1 | 23.3 | 8.6 | 32.0 | 41.3 |
| + SFT + RL ($\alpha = 0.25$) | 36.1 | 53.0 | 55.1 | 33.3 | 46.3 | 34.3 | 36.2 | 40.3 | 17.3 | 29.3 | 7.0 | 35.2 | 45.8 |
| + SFT + RL ($\alpha = 0.50$) | 35.8 | 51.3 | 54.3 | 32.7 | 47.4 | 31.7 | 35.0 | 39.3 | 20.7 | 29.2 | 8.2 | 34.6 | 44.6 |
| + SFT + RL ($\alpha = 0.75$) | 36.6 | 53.7 | 55.9 | 35.6 | 47.9 | 34.0 | 35.6 | 39.3 | 17.6 | 29.8 | 9.6 | 35.4 | 45.0 |

that our reasoning-enhanced pointwise approach is general-purpose and suitable for both reasoning-intensive and traditional retrieval scenarios. Moreover, by integrating over 16 sampling scores, the model achieves a final nDCG@10 of 56.8, indicating the effectiveness of test-time scaling even on traditional benchmarks, as illustrated in Figure 6.

## F ANALYSIS OF THE HYPERPARAMETER $\alpha$

Table 9 reports the performance of training with different weights $\alpha$ for the composite rewards. When $\alpha = 0$ or $\alpha = 1$, the model degenerates into using only one of the composite rewards, resulting in suboptimal performance. In particular, training with only the intra-document reward ($\alpha = 1$) clearly outperforms training with only the inter-document reward ($\alpha = 0$), suggesting that the intra-reward provides a stronger optimization signal for model learning. This observation aligns with its role in stabilizing training by encouraging more consistent reasoning trajectories within a document.

However, despite its dominance, the intra-reward alone still underperforms any weighted combination of the two rewards. Across all $\alpha \in (0, 1)$, we observe consistent performance gains over using either reward alone, demonstrating that the intra- and inter-document rewards serve complementary purposes. While the intra-reward promotes stable and learnable ranking behavior, the inter-reward introduces document-level discrimination that further enhances final ranking performance.

These results indicate that effective reward balancing enables the model to simultaneously exploit stable intra-document ranking signals and informative inter-document comparisons. Notably, performance peaks when $\alpha$ is biased toward the intra-reward (e.g., $\alpha = 0.75$), highlighting that the intra-reward should dominate training dynamics, while the inter-reward acts as an essential complementary signal that enhances ranking performance.

## G DIFFERENT BACKBONE MODELS

To demonstrate the generalizability and effectiveness of our Retro*, we conducted experiments with several different backbone models. As shown in Table 10, Retro* significantly improves the average nDCG@10 across all backbone models: Qwen2.5-Instruct (7B) from 22.9 to 36.6, Llama2-chat (7B) from 8.2 to 30.9, Mistral-Instruct-v0.1 (7B) from 8.6 to 30.1, Llama3.1-Instruct (8B) from 17.3 to 34.4, and Qwen3 (8B) from 29.4 to 36.1. Furthermore, applying test-time scaling further improves performance consistently for all backbones as shown in Figure 6. These results demonstrate that our approach consistently and substantially enhances ranking performance across diverse backbones, while test-time scaling remains effective with each backbone, further highlighting its robustness and broad applicability as a general retrieval model.

## H ANALYSIS OF THE LENGTH CONTROL INSTRUCTION FOR DISTILLATION

In this section, we analyze the impact of the length control instruction used during the distillation from the teacher model. We augment the prompt template (shown in Figure 8) with an explicit instruction at the beginning of the task description. The modified prompt template is shown below:

```
Here is the **relevance definition** in a retrieval task: {relevance_definition}
```

Table 10: nDCG@10 results on the BRIGHT benchmark across different backbone models. All backbone models re-rank the top-100 documents retrieved by BGE-Reasoner-Embed. The retrieval stage and the re-ranking stage both utilize the original query.

| Models | Avg. | StackExchange | | | | | | | Coding | | Theorem-based | | |
|---|---|---|---|---|---|---|---|---|---|---|---|---|---|
| | | Bio. | Earth. | Econ. | Psy. | Rob. | Stack. | Sus. | Leet. | Pony | AoPS | TheoQ. | TheoT. |
| BGE-Reasoner-Embed | 32.5 | 42.6 | 42.6 | 27.8 | 37.3 | 26.4 | 29.6 | 30.6 | 36.9 | 25.7 | 9.8 | 34.9 | 46.1 |
| Qwen2.5-Instruct (7B) | 22.9 | 39.9 | 41.2 | 21.0 | 31.4 | 17.0 | 16.9 | 22.7 | 12.1 | 15.7 | 3.9 | 14.2 | 38.7 |
| Retro* (7B) | 36.6 | 53.7 | 55.9 | 35.6 | 47.9 | 34.0 | 35.6 | 39.3 | 17.6 | 29.8 | 9.6 | 35.4 | 45.0 |
| *Test-Time Scaling (Mean-Score@16)* | | | | | | | | | | | | | |
| Retro* (7B) | 38.7 | 58.4 | 59.2 | 35.0 | 49.3 | 33.9 | 37.7 | 41.1 | 18.8 | 33.5 | 10.7 | 40.2 | 46.7 |
| Llama2-chat (7B) | 8.2 | 13.4 | 14.6 | 8.2 | 11.2 | 7.0 | 7.6 | 10.6 | 7.6 | 4.4 | 2.4 | 3.2 | 8.7 |
| Retro* (7B) | 25.4 | 43.7 | 40.1 | 22.0 | 35.8 | 20.7 | 26.8 | 29.7 | 18.1 | 19.4 | 6.6 | 22.0 | 19.5 |
| *Test-Time Scaling (Mean-Score@16)* | | | | | | | | | | | | | |
| Retro* (7B) | 30.9 | 53.3 | 49.6 | 28.8 | 42.8 | 25.7 | 29.9 | 34.3 | 22.1 | 25.1 | 7.5 | 29.2 | 23.1 |
| Mistral-Instruct-v0.1 (7B) | 8.6 | 17.2 | 15.4 | 6.9 | 11.6 | 7.7 | 8.3 | 10.6 | 6.0 | 7.7 | 2.5 | 3.8 | 5.5 |
| Retro* (7B) | 27.8 | 49.1 | 42.4 | 24.4 | 38.0 | 23.7 | 29.7 | 31.9 | 15.5 | 20.7 | 9.3 | 24.5 | 24.5 |
| *Test-Time Scaling (Mean-Score@16)* | | | | | | | | | | | | | |
| Retro* (7B) | 30.1 | 53.9 | 46.0 | 27.3 | 43.0 | 25.7 | 30.9 | 31.6 | 15.0 | 24.0 | 10.1 | 27.8 | 26.1 |
| Llama3.1-Instruct (8B) | 17.3 | 35.0 | 27.8 | 17.3 | 30.0 | 11.8 | 19.5 | 21.3 | 8.8 | 4.8 | 4.2 | 11.4 | 15.1 |
| Retro* (8B) | 34.4 | 56.7 | 53.5 | 31.3 | 47.3 | 31.3 | 32.7 | 40.2 | 19.9 | 21.2 | 9.4 | 33.0 | 36.4 |
| *Test-Time Scaling (Mean-Score@16)* | | | | | | | | | | | | | |
| Retro* (8B) | 36.7 | 60.2 | 55.9 | 33.5 | 49.7 | 33.6 | 36.2 | 42.0 | 19.0 | 24.7 | 10.3 | 38.4 | 36.4 |
| Qwen3 (8B) | 29.4 | 53.5 | 53.8 | 28.3 | 34.1 | 26.3 | 29.2 | 33.1 | 14.1 | 8.4 | 6.7 | 20.6 | 44.4 |
| Retro* (8B) | 36.1 | 55.0 | 56.2 | 34.9 | 42.0 | 35.9 | 35.6 | 42.5 | 16.7 | 21.9 | 8.4 | 35.4 | 48.7 |
| *Test-Time Scaling (Mean-Score@16)* | | | | | | | | | | | | | |
| Retro* (8B) | 38.8 | 57.6 | 58.5 | 37.1 | 45.8 | 38.3 | 39.3 | 45.4 | 17.3 | 26.6 | 10.6 | 40.4 | 48.8 |

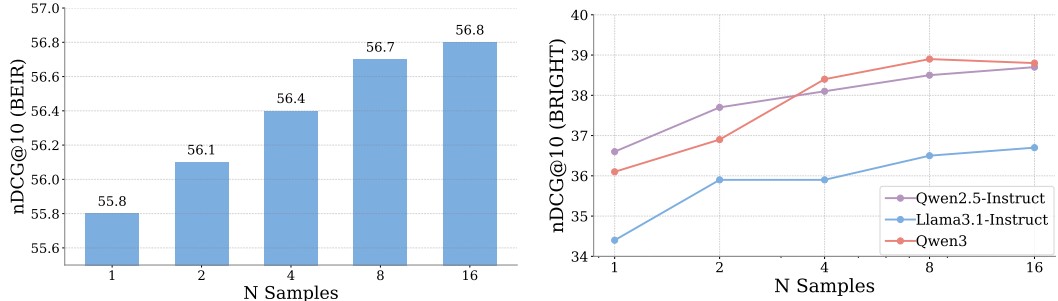

Figure 6: Effectiveness of Test-Time Scaling for Retro*. **(Left):** Test-time scaling brings significant performance improvements even on the traditional BEIR benchmark. **(Right):** Test-time scaling is consistently effective across different backbone models.

```
Now given a **query** ({query_type}) and a **document** ({doc_type}) in this retrieval task,
    your mission is to perform the following steps (**Please ensure your entire analysis and
    annotation across all steps does not exceed 512 tokens**).

... (The rest of the prompt template remains the same)
```

To evaluate the effect of this instruction on both the reasoning length and the model's performance, we construct two distillation datasets from the teacher model under two conditions: one with the instruction and one without it. We then train student models on these datasets, respectively. The impact on reasoning length is illustrated in Figure 7, while the model performance is reported in Table 11.

As shown in Figure 7, the length control instruction substantially shortens the reasoning trajectories, yielding an average trajectory length comparable to that of the backbone model. Meanwhile, Table 11 shows that the model trained with this instruction achieves an average nDCG@10 of 36.6, essentially matching the 36.4 achieved without it. These results demonstrate that the instruction effectively constrains the reasoning length for efficiency without compromising re-ranking accuracy. Furthermore, although the reasoning length of our trained model is close to that of the backbone,

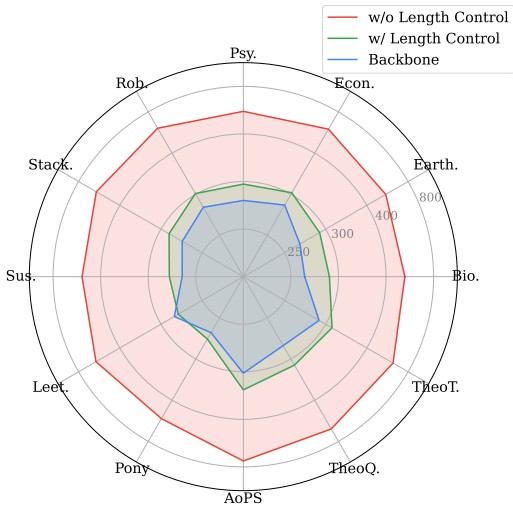

Figure 7: Comparison of average completion tokens on BRIGHT benchmark for different models. The model trained with the length control instruction (w/ Length Control) effectively reduces the response length compared to training without it (w/o Length Control).

Table 11: nDCG@10 results on the BRIGHT benchmark for ablation study on the impact of the length control instruction.

| Models | Avg. | StackExchange | | | | | | | Coding | | Theorem-based | | |
| --- | --- | --- | --- | --- | --- | --- | --- | --- | --- | --- | --- | --- | --- |
| | | Bio. | Earth. | Econ. | Psy. | Rob. | Stack. | Sus. | Leet. | Pony | AoPS | TheoQ. | TheoT. |
| BGE-Reasoner-Embed | 32.5 | 42.6 | 42.6 | 27.8 | 37.3 | 26.4 | 29.6 | 30.6 | 36.9 | 25.7 | 9.8 | 34.9 | 46.1 |
| Qwen2.5-Instruct (7B) | 22.9 | 39.9 | 41.2 | 21.0 | 31.4 | 17.0 | 16.9 | 22.7 | 12.1 | 15.7 | 3.9 | 14.2 | 38.7 |
| *w/ Length Control* | | | | | | | | | | | | | |
| + only-SFT | 30.1 | 46.9 | 51.3 | 29.1 | 37.4 | 24.4 | 28.4 | 35.0 | 35.0 | 20.6 | 7.9 | 27.9 | 36.6 |
| + SFT + RL (Composite Reward) | 36.6 | 53.7 | 55.9 | 35.6 | 47.9 | 34.0 | 35.6 | 39.3 | 17.6 | 29.8 | 9.6 | 35.4 | 45.0 |
| *w/o Length Control* | | | | | | | | | | | | | |
| + only-SFT | 30.7 | 46.8 | 50.2 | 30.5 | 39.4 | 25.7 | 30.1 | 35.8 | 14.4 | 18.0 | 7.6 | 27.9 | 41.6 |
| + SFT + RL (Composite Reward) | 36.4 | 53.3 | 55.6 | 33.7 | 48.7 | 33.3 | 35.6 | 39.5 | 16.5 | 31.1 | 8.3 | 36.4 | 45.1 |

it exhibits stronger ranking performance, suggesting that the improvement does not arise from generating longer reasoning trajectories. Instead, it stems from our effective training strategy, which equips the model with powerful reasoning and ranking capabilities.

# I  OUT-OF-DOMAIN GENERALIZATION ON R2MED

To evaluate the out-of-domain generalizability of our rubric, we evaluate Retro* on R2MED (Li et al., 2025), a biomedical reasoning retrieval benchmark that is outside the scientific, programming, and mathematical domains used during training. By replacing only the task-specific relevance definitions while keeping the task-agnostic scoring criteria, Retro* is able to generalize effectively to this unseen domain. Following the setup of ReasonRank, we use E5-mistral-7b-instruct (Wang et al., 2024) as the first-stage retriever, re-rank the top-100 retrieved candidates, and report nDCG@10. As summarized in Table 12, Retro* outperforms all other re-ranking baselines, demonstrating both the robustness of its rubric-based scoring and its strong cross-domain generalization capability.

# J  RUBRICS OF RUBRIC-BASED RELEVANCE SCORING

We provide the full relevance rubrics for our rubric-based relevance scoring mechanism in Figure 8. The rubrics are designed to guide the LLMs to reason about the relevance between a query and a candidate document through a three-step reasoning process (query analysis, document analysis, and relevance annotation) to produce a fine-grained relevance score.

```
Here is the **relevance definition** in a retrieval task: {relevance_definition}

Now given a **query** ({query_type}) and a **document** ({doc_type}) in this retrieval task,
    your mission is to perform the following steps.

1. Query Analysis: Think to reason and describe what information would be most helpful in
    answering the query.
2. Document Analysis: Discuss how the information provided by the document fulfills or fails
    to fulfill the requirements implied by the query.
3. Relevance Annotation: Based on the relevance definition and the insights from the previous
    two steps, clearly justify your final relevance annotation result and annotate an integer
    score from a scale of 0 to 100. Please use the following guide:
   - **80-100 (Highly Relevant):** The document directly and comprehensively addresses the
       query's intent. It is a core and authoritative answer.
   - **60-80 (Relevant):** The document substantially addresses the query's intent, providing
       most of the key information, but might miss some minor details.
   - **40-60 (Moderately Relevant):** The document is on-topic and addresses a part of the
       query's intent, but it is not a comprehensive answer.
   - **20-40 (Slightly Relevant):** The document mentions keywords from the query, but its
       main topic is different. It offers very limited value.
   - **0-20 (Irrelevant):** The document does not address the query's intent at all and is
       off-topic.

After providing your detailed analysis and justification for all the steps above, conclude
    your entire response with the final relevance score. The score must be placed strictly
    between the <score> tags. There should be no other text or explanation inside the tags:
<score>
[From a scale of 0 to 100, annotate the degree of relevance between the query and the document
    .]
</score>

Query ({query_type}):
[Begin of Query]
{query}
[End of Query]

Document ({doc_type}):
[Begin of Document]
{doc}
[End of Document]
```

Figure 8: The full relevance rubrics for rubric-based relevance scoring.

Table 12: nDCG@10 results on the R2MED benchmark, where all methods re-rank the top-100 documents retrieved by E5-mistral-7b-instruct. The retrieval stage and the re-ranking stage both utilize the original query. Results marked with [†] are reported by ReasonRank.

| Models | Methods | Avg. | Biolo. | Bioin. | Med-Sci. | Med-Exam. | Med-Diag. | PMC-Treat. | PMC-Cli. | IIYi-Cli. |
|---|---|---|---|---|---|---|---|---|---|---|
| E5-mistral-7b-instruct | Retriever | 24.0 | 22.9 | 42.3 | 41.5 | 7.4 | 12.4 | 18.5 | 24.9 | 21.8 |
| *Reasoning-Enhanced Re-Ranking Baselines* | | | | | | | | | | |
| Rank1[†] (7B) | Pointwise | 32.3 | 32.6 | 55.6 | 54.7 | 12.8 | 20.0 | 34.4 | 30.2 | 18.2 |
| Rank1[†] (32B) | Pointwise | 39.1 | 31.8 | 61.7 | 59.7 | 16.6 | 26.9 | 41.3 | 45.6 | 29.5 |
| ReasonRank[†] (7B) | Listwise | 39.5 | 46.8 | 59.7 | 60.1 | 16.5 | 24.9 | 39.2 | 39.1 | 29.9 |
| ReasonRank[†] (32B) | Listwise | 42.9 | 45.6 | 67.7 | 63.5 | 18.9 | 30.6 | 41.1 | 46.1 | 29.4 |
| Retro* (7B) | Pointwise | 38.7 | 47.6 | 59.7 | 60.8 | 12.7 | 25.2 | 40.7 | 39.0 | 24.2 |
| Retro* (32B) | Pointwise | 44.5 | 48.8 | 68.0 | 67.3 | 16.6 | 34.2 | 43.3 | 45.4 | 32.6 |
| *Test-Time Scaling (Mean-Score@16)* | | | | | | | | | | |
| Retro* (7B) | Pointwise | 42.6 | 49.9 | 62.4 | 62.4 | 15.7 | 29.0 | 41.9 | 46.2 | 32.9 |
| Retro* (32B) | Pointwise | 46.2 | 51.9 | 67.8 | 67.2 | 19.2 | 35.7 | 43.8 | 48.7 | 35.6 |

## K ANALYSIS OF THE NECESSITY OF REASONING

To investigate the necessity of reasoning for re-ranking performance, we conduct an ablation study by removing the structured reasoning steps from our relevance rubrics. In the original rubrics, the model is prompted to analyze both the query and the candidate document before producing a detailed relevance score. In the ablated variant, the model instead outputs a single score directly, without any reasoning trajectory or rubric-guided analysis, as illustrated in Figure 9.

Table 13: Relevance definitions for each dataset in the BRIGHT benchmark.

| Dataset | Relevance Definition |
|---|---|
| biology | Given a query (biology post) and a document (passage), the document is relevant to the query if the critical concepts or theories discussed in the document can provide references for domain experts to draft an answer to the query. |
| earth_science | Given a query (earth science post) and a document (passage), the document is relevant to the query if the critical concepts or theories discussed in the document can provide references for domain experts to draft an answer to the query. |
| economics | Given a query (economics post) and a document (passage), the document is relevant to the query if the critical concepts or theories discussed in the document can provide references for domain experts to draft an answer to the query. |
| psychology | Given a query (psychology post) and a document (passage), the document is relevant to the query if the critical concepts or theories discussed in the document can provide references for domain experts to draft an answer to the query. |
| robotics | Given a query (robotics post) and a document (passage), the document is relevant to the query if the critical concepts or theories discussed in the document can provide references for domain experts to draft an answer to the query. |
| stackoverflow | Given a query (Stack Overflow post) and a document (passage), the document is relevant to the query if the critical concepts or theories discussed in the document can provide references for domain experts to draft an answer to the query. |
| sustainable_living | Given a query (sustainable living post) and a document (passage), the document is relevant to the query if the critical concepts or theories discussed in the document can provide references for domain experts to draft an answer to the query. |
| leetcode | Given a query (LeetCode problem) and a document (coding problem solution), the document is relevant to the query if the underlying algorithms or data structures used in the document can provide helpful insights for solving the problem in the query. |
| pony | Given a query (Pony coding instruction) and a document (Pony documentation passage), the document is relevant to the query if the Pony syntax described in the document is necessary for beginners with no prior knowledge of Pony to complete the coding instruction in the query. |
| aops | Given a query (math problem) and a document (math problem solution), the document is relevant to the query if the theorems used in the document can provide helpful insights for solving the problem in the query. |
| theoremqa_questions | Given a query (math problem) and a document (math problem solution), the document is relevant to the query if the theorems used in the document can provide helpful insights for solving the problem in the query. |
| theoremqa_theorems | Given a query (math problem) and a document (math-related passage), the document is relevant to the query if the theorem described in the document can help solve the problem in the query. |

This variant of Retro* is trained on the same dataset and with the same RL algorithm as the original model. We focus the comparison specifically on the RL stage to avoid introducing confounding factors, such as potential biases originating from the SFT teacher model. By keeping the learning signal consistent, this setup effectively isolates the impact of reasoning on re-ranking performance. As shown in Table 16, removing the reasoning trajectory results in a substantial performance drop across all datasets. These results confirm that rubric-guided reasoning enhances the fidelity of relevance estimation under RL training. By structuring the task into intermediate steps, the model can better capture how different aspects of relevance contribute to the final judgment. Without this reasoning guidance, the model struggles to learn the underlying compositional logic of the relevance rubric, leading to the observed performance drop.

## L    TASK RELEVANCE DEFINITIONS FOR DIFFERENT BENCHMARKS

In this section, we provide the detailed relevance definitions used in our evaluations on the BRIGHT, BEIR and R2MED benchmarks. In reasoning-intensive retrieval scenarios, the relevance definitions are more complex, while in traditional scenarios, they are more straightforward. The specific defi-

Table 14: Relevance definitions for each dataset in the BEIR benchmark.

| Dataset | Relevance Definition |
|---------|---------------------|
| TREC-COVID | Given a query (COVID-19 related query) and a document (document), the document is relevant to the query if the document answers the query. |
| DBPedia | Given a query (query) and a document (entity description from DBpedia), the document is relevant to the query if the entity described in the document matches the query. |
| SciFact | Given a query (scientific claim) and a document (document), the document is relevant to the query if the document provides evidence supporting or refuting the scientific claim. |
| NFCorpus | Given a query (question) and a document (document), the document is relevant to the query if the document can best answer the question. |
| Signal-1M | Given a query (news event or topic) and a document (news headline or summary), the document is relevant to the query if it reports on, summarizes, or directly relates to the same news event or topic described in the query. |
| Robust04 | Given a query (information need) and a document (news or government document), the document is relevant to the query if it contains information that satisfies the intent or topic described in the query, even if phrased differently. |
| TREC-NEWS | Given a query (contemporary news topic or event) and a document (news article from The Washington Post), the document is relevant to the query if it discusses, explains, or provides factual coverage of the specific event or topic mentioned in the query. |

Table 15: Relevance definitions for each dataset in the R2MED benchmark.

| Dataset | Relevance Definition |
|---------|---------------------|
| Biology | Given a query (biology post) and a document (passage), the document is relevant to the query if the document helps answer the query. |
| Medical-Sciences | Given a query (medical science post) and a document (passage), the document is relevant to the query if the document helps answer the query. |
| Bioinformatics | Given a query (bioinformatics post) and a document (passage), the document is relevant to the query if the document helps answer the query. |
| MedXpertQA-Exam | Given a query (medical exam) and a document (passage), the document is relevant to the query if the document helps answer the query. |
| MedQA-Diag | Given a query (medical exam) and a document (passage), the document is relevant to the query if the document helps answer the query. |
| PMC-Treatment | Given a query (clinical case) and a document (passage), the document is relevant to the query if the document helps answer the query. |
| PMC-Clinical | Given a query (clinical case) and a document (case), the document is relevant to the query if it helps diagnose the query case. |
| IIYi-Clinical | Given a query (clinical case) and a document (case), the document is relevant to the query if it helps diagnose the query case. |

Table 16: Performance comparison on the BRIGHT benchmark, showing that reasoning-enhanced re-ranking significantly outperforms models trained without reasoning.

| Models | Avg. | StackExchange | | | | | | | Coding | | Theorem-based | | |
|--------|------|------|-------|------|------|------|--------|------|-------|------|------|-------|--------|
| | | Bio. | Earth. | Econ. | Psy. | Rob. | Stack. | Sus. | Leet. | Pony | AoPS | TheoQ. | TheoT. |
| BGE-Reasoner-Embed | 32.5 | 42.6 | 42.6 | 27.8 | 37.3 | 26.4 | 29.6 | 30.6 | 36.9 | 25.7 | 9.8 | 34.9 | 46.1 |
| No Reason (only-RL) | 24.8 | 38.4 | 40.2 | 22.3 | 32.4 | 20.5 | 25.6 | 26.0 | 19.6 | 16.9 | 4.7 | 17.7 | 33.8 |
| Reason (only-RL) | 35.1 | 56.7 | 52.9 | 33.2 | 45.1 | 28.1 | 32.5 | 35.8 | 21.2 | 36.3 | 8.6 | 29.8 | 41.0 |

nitions for the BRIGHT benchmark are listed in Table 13, BEIR are listed in Table 14, and those for R2MED are listed in Table 15. We use these relevance definitions for both training and evaluation.

# M    REASONING EXAMPLES

We provide some examples to present the reasoning process of Retro* in Table 17 (Suatainable Living), Table 18 (Pony), and Table 19 (TheoremQA Theorems).

```
Here is the **relevance definition** in a retrieval task: {relevance_definition}

Now given a **query** ({query_type}) and a **document** ({doc_type}) in this retrieval task,
    your mission is to perform the following steps.

Based on the relevance definition, annotate an integer score from a scale of 0 to 100. Please
    use the following guide: 80-100 (Highly Relevant), 60-80 (Relevant), 40-60 (Moderately
    Relevant), 20-40 (Slightly Relevant), 0-20 (Irrelevant).

Without any explanation, directly providing your final relevance score. The score must be
    placed strictly between the <score> tags. There should be no other text or explanation
    inside the tags:
<score>
[From a scale of 0 to 100, annotate the degree of relevance between the query and the document
    .]
</score>

Query ({query_type}):
[Begin of Query]
{query}
[End of Query]

Document ({doc_type}):
[Begin of Document]
{doc}
[End of Document]
```

Figure 9: The modified relevance rubrics for reasoning-free relevance scoring.

Table 17: Reasoning examples of Retro* (7B) on the **Sustainable Living** dataset of BRIGHT.

---

**Relevance Definition:** Given a query (sustainable living post) and a document (passage), the document is relevant to the query if the critical concepts or theories discussed in the document can provide references for domain experts to draft an answer to the query.

---

**Query:**
Are solar powered cars sustainable and are they a good idea for future transportation?\n \nAs population rises, fossil fuels will start depleting so are solar powered cars a good idea? How much power can they produce when it is cloudy?

---

**Positive Document:**
The efficiency of the solar cells used in a [ photovoltaic system\n ](/wiki/Photovoltaic_system "Photovoltaic system") , in combination with\nlatitude and climate, determines the annual energy output of the system. For\nexample , a solar panel with 20% efficiency and an area of 1 m 2 will produce\n 200 kWh/yr at Standard Test Conditions if exposed to the Standard Test\nCondition solar irradiance value of 1000 W/m 2 for 2.74 hours a day. Usually\nsolar panels are exposed to sunlight for longer than this in a given day, but\nthe solar irradiance is less than 1000 W/m 2 for most of the day. A solar\npanel can produce more when the Sun is high in Earth's sky and will produce\nless in cloudy conditions or when the Sun is low in the sky, usually the Sun\nis lower in the sky in the winter.
**Reasoning Process:**
1. Query Analysis: The query seeks an evaluation of solar-powered cars in terms of sustainability and their viability as future transportation. It specifically asks whether solar cars are sustainable and produce usable power under cloudy conditions. A helpful response would discuss solar car efficiency, energy capture, potential as a sustainable transport solution, and the impact of weather conditions on system output.

2. Document Analysis: The passage explains how solar panel efficiency, climate, and solar patterns affect energy output, particularly noting reduced power under cloudy skies and low-sunlight conditions. While it validates solar panel performance under varying light conditions—directly relevant to solar car feasibility—it does not mention solar-powered cars, automotive application, transportation systems, or sustainability over time. However, it provides foundational data on power generation under cloudiness, addressing part of the query.

3. Relevance Annotation: The document supplies critical information about solar panel energy output affected by cloudiness, which is pertinent to evaluating solar-powered cars. However, it does not link this to transportation solutions, sustainability assessment of solar cars, or holistic future outlook. It offers technical underpinnings but lacks domain-specific context regarding vehicles or long-term strategies. Thus, while informative for part of the query (especially on power availability in cloudy conditions), it omits major integrative insights needed for a comprehensive answer.

<score>
65
</score>

---

**Negative Document:**
The first solar family car was built in 2013. [16] Researchers at [ Case\nWestern Reserve University ](/wiki/Case_Western_Reserve_University "Case\nWestern Reserve University") , have also developed a solar car which can\nrecharge more quickly, due to materials used in the solar panels. [17]
**Reasoning Process:**
1. Query Analysis: The query seeks to evaluate the sustainability and practicality of solar-powered cars, particularly in the context of future transportation. It emphasizes two key aspects: (1) whether they are sustainable (e.g., environmental impact, energy efficiency), and (2) whether they are a good idea for addressing fossil fuel depletion and scalability, especially when sunlight availability fluctuates under clouds. Domain experts would need information on energy output, charging capacity, environmental benefits, and real-world feasibility.

2. Document Analysis: The document mentions a solar car from 2013 and references research improving charging speed with advanced materials. However, it lacks any discussion of sustainability, energy production, energy cloud dependency, or broader transportation implications. It provides only a limited technical note on material advancements without addressing the core sustainability or societal questions raised.

3. Relevance Annotation: While the document is tangentially related—mentioning solar-powered cars—it fails to address the critical concepts in the query, such as sustainability metrics, energy output under cloudy conditions, or long-term viability. It does not offer references that would help draft a comprehensive answer. The information is too minimal and outdated to support expert judgment on the broader query.

<score>
35
</score>

---

Table 18: Reasoning examples of Retro* (7B) on the **Pony** dataset of BRIGHT.

---

**Relevance Definition:** Given a query (Pony coding instruction) and a document (Pony documentation passage), the document is relevant to the query if the Pony syntax described in the document is necessary for beginners with no prior knowledge of Pony to complete the coding instruction in the query.

---

**Query:**
I will use the programming language pony.\nProblem :\nWrite a function to determine if a string consisting of alphabets is a palindrome.\n \nHere is the code template:\nfun is_palindrome(str: String): Bool ? =>\n ...

---

**Positive Document:**
## Functions\n \nPony functions are quite like functions (or methods) in other languages. They can have 0 or more parameters and 0 or 1 return values. If the return type is omitted then the function will have a return value of 'None'.\n \n "'pony\nclass C\nfun add(x: U32, y: U32): U32 =>\nx + y\n \nfun nop() =>\nadd (1, 2) // Pointless, we ignore the result\n "'\n \nThe function parameters (if any) are specified in parentheses after the function name. Functions that don't take any parameters still need to have the parentheses.\n \nEach parameter is given a name and a type. In our example function 'add' has 2 parameters, 'x' and 'y', both of which are type 'U32'. The values passed to a function call (the '1' and '2' in our example) are called arguments and when the call is made they are evaluated and assigned to the parameters. Parameters may not be assigned to within the function - they are effectively declared 'let'.\n \nAfter the parameters comes the return type. If nothing will be returned this is simply omitted.\n \nAfter the return value, there's a '=>' and then finally the function body. The value returned is simply the value of the function body (remember that everything is an expression), which is simply the value of the last command in the function.

**Reasoning Process:**
1. Query Analysis:
The query asks to write a function in Pony that determines whether a string is a palindrome. The key elements needed are: (1) a function taking a parameter of type 'String' and returning a 'Bool', (2) string handling in Pony, (3) string reversal/iteration, and (4) comparison logic. However, for a beginner with no prior knowledge, the most critical information is understanding how to define a function in Pony-especially the syntax for defining a function with a parameter, return type, and body. This is foundational for structuring any function, including one for palindrome detection.

2. Document Analysis:
The document explains Pony function syntax in depth, covering parameter specification, return types (including optional ones), function bodies, and return values using a basic example ('add'). It clarifies that functions use 'fun', can have parameter lists with types, return statements via evaluation of the final expression, and can omit return types for 'None'. While it does not mention string operations, it provides essential syntax needed to write any function, including 'is_palindrome'.

3. Relevance Annotation:
The document does not address handling strings or palindrome logic directly, but it fully covers the necessary syntactic structure for defining the function in the query-parameterized functions with return types, the 'fun' statement, and function bodies. This is foundational knowledge a beginner must have to implement the requested function. While not comprehensive for the full task, it supplies the core syntactic framework. Thus, it is highly relevant to the coding instruction in terms of necessary beginner syntax.

<score>
75
</score>

---

**Negative Document:**
## Control Structures\n \nTo do real work in a program you have to be able to make decisions, iterate through collections of items and perform actions repeatedly. For this, you need control structures. Pony has control structures that will be familiar to programmers who have used most languages, such as 'if', 'while' and 'for', but in Pony, they work slightly differently.\n

**Reasoning Process:**
1. Query Analysis: The query asks for a function in Pony to check if a string is a palindrome. This requires understanding of string manipulation, comparison, and possibly loops or recursion. Key concepts for beginners include: defining a function with the correct signature (e.g., 'String → Bool'), accessing characters in a string, reverse comparison, and possibly handling case sensitivity or ignoring non-alphabetic characters. The critical syntax needed includes function definition, type parameters, string indexing, and comparison logic.

2. Document Analysis: The document discusses general programming control structures ('if', 'while', 'for') and notes that these work similarly in Pony. However, it provides no specific examples, syntax, or details on string handling, function syntax, or palindrome logic. It fails to mention how to manipulate strings, access characters, or implement palindrome detection-core needs for the query. The content is introductory and abstract, focusing on high-level structure rather than concrete, applicable code.

3. Relevance Annotation: The document touches on conditional and iterative constructs, which are tangential to palindrome logic, but it does not provide the necessary syntax or examples for completing the task. It lacks any mention of string properties, indexing, or function definition relevant to the query. For a beginner, this passage would offer minimal direct assistance. Therefore, it only shares conceptual keywords (e.g., "if", "while") but misses all domain-specific syntax.

<score>
5
</score>

---

Table 19: Reasoning examples of Retro* (7B) on the **TheoremQA Theorems** dataset of BRIGHT.

**Relevance Definition:** Given a query (math problem) and a document (math-related passage), the document is relevant to the query if the theorem described in the document can help solve the problem in the query.

**Query:**
In a party, how many guests do you need to have to ensure that either four people all know each other or four people are all complete strangers to one another?

**Positive Document:**
\\section{Ramsey's Theorem}\nTags : Ramsey Theory, Named Theorems, Combinatorics\n \n \\begin{theorem}\nIn any coloring of the edges of a sufficiently large complete graph, one will find monochromatic complete subgraphs.\nFor 2 colors, Ramsey's theorem states that for any pair of positive integers $\\tuple {r, s}$, there exists a least positive integer $\\map R {r, s}$ such that for any complete graph on $\\map R {r, s}$ vertices, whose edges are colored red or blue, there exists either a complete subgraph on $r$ vertices which is entirely red, or a complete subgraph on $s$ vertices which is entirely blue.\nMore generally, for any given number of colors $c$, and any given integers $n_1, \\ldots, n_c$, there is a number $\\map R {n_1, \\ldots, n_c}$ such that:\n :if the edges of a complete graph of order $\\map R {n_1, \\ldots, n_c}$ are colored with $c$ different colours, then for some $i$ between $1$ and $c$, it must contain a complete subgraph of order $n_i$ whose edges are all color $i$.\nThis number $\\map R {n_1, \\ldots, n_c}$ is called the Ramsey number for $n_1, \\ldots, n_c$.\nThe special case above has $c = 2$ (and $n_1 = r$ and $n_2 = s$).\nHere $\\map R {r, s}$ signifies an integer that depends on both $r$ and $s$. It is understood to represent the smallest integer for which the theorem holds.\n \\end{theorem}\n \n \\begin{proof}\nFirst we prove the theorem for the 2-color case, by induction on $r + s$.\nIt is clear from the definition that\n :$\\forall n \\in \\N: \\map R {n, 1} = \\map R {1, n} = 1$\nbecause the complete graph on one node has no edges.\nThis is the base case.\nWe prove that $R \\left({r, s}\\right)$ exists by finding an explicit bound for it.\nBy the inductive hypothesis, $\\map R {r - 1, s}$ and $\\map R {r, s - 1}$ exist.\n \\end{proof}\n \n

**Reasoning Process:**
1. Query Analysis: The query asks for the minimum number of guests (people) in a party such that either 4 people all know each other (a complete subgraph $K_4$ in a "knows"-graph) or 4 people are all strangers to each other (a complete subgraph $K_4$' in a "strangers"-graph). This is a classic Ramsey number problem, specifically asking for $R(4,4)$, the smallest number of guests such that in any 2-coloring of the edges (representing know-or-stranger), there must be a monochromatic $K_4$. The key concept needed is Ramsey theory, particularly the Ramsey number $R(4,4)$.

2. Document Analysis: The document presents Ramsey's Theorem, defining the Ramsey number $\map R {r,s}$ as the smallest number such that any 2-coloring of the edges of a complete graph on $\map R {r,s}$ vertices guarantees either an all-red $K_r$ or all-blue $K_s$. It confirms the theorem for $c=2$, which directly applies to the query's 2-coloring framework. While it does not explicitly state $R(4,4)$ or compute its value (which is 18), the theorem provides the theoretical foundation to solve the query. From this, one could deduce that any graph on 18 or more vertices necessarily contains either a $K_4$ or $\overline {K}_4$, depending on the coloring.

3. Relevance Annotation: The document is highly relevant because it introduces and explains Ramsey's Theorem, the core mathematical principle required to solve the query. Although it does not provide the specific value of $R(4,4)$, it enables one to derive it via the theorem. It directly supports solving the problem by offering the theoretical framework.

<score>
95
</score>

**Negative Document:**
\\section{Infinite Ramsey's Theorem}\nTags : Ramsey Theory, Named Theorems\n \n \\begin{theorem}\nLet $k, n \\in \\N$.\nFor any set $S$, let $S^{\\paren n}$ denote the set $\\set {\\set {s_1, \\ldots, s_n}: \\text{each } s_i \\in S}$ of cardinality $n$ subsets of $S$.\nLet $X$ be an infinite set.\nThen :\n :for every partition $P$ of $X^{\\paren n}$ into $k$ many components\n :there is an infinite subset $Y \\subseteq X$\nsuch that:\n :each member of $Y^{\\paren n}$ is in the same component of $P$.\n \\end{theorem}\n \n \\begin{proof}\nWe will prove the theorem for fixed $k$ by induction on $n$.\n \\end{proof}\n \n

**Reasoning Process:**
1. Query Analysis: The query asks for the minimum number of guests (n) at a party such that either four people all know each other (a clique of size 4 in a social graph) or four mutually unknown people exist (an independent set of size 4). This is a Ramsey theory problem specifically involving the Ramsey number R(4,4), which is the smallest number n such that any 2-coloring of the edges of a complete graph on n vertices guarantees a monochromatic clique of size 4 or an independent set of size 4. The key information needed is a known result about R(4,4), preferably its value or a discussion of its combinatorial implications.

2. Document Analysis: The document presents the Infinite Ramsey Theorem, which states that for any partition of the n-element subsets of an infinite set into finitely many components, there exists an infinite subset that monochromatic. This is a much more general and abstract result. While it is situated in Ramsey theory, it deals with infinite sets and n-element subsets, and does not address the finite Ramsey number R(4,4) or any finite Ramsey-type problem. It does not mention graph theory, social networks, cliques, or independent sets in the context of a finite party problem. Thus, it fails to provide any specific insight into the numerical or combinatorial condition required to solve the query.

3. Relevance Annotation: The document is thematically related (Ramsey Theory) but addresses a fundamentally different domain (infinite sets, infinite Ramsey Theorem) and offers no useful method or result for determining R(4,4) or solving the finite problem. It cannot help directly in solving the given social puzzle. While both involve Ramsey theory, the connection is too abstract and generic without addressing the specific problem or its mathematical tools.

<score>
10
</score>

