# OpenReview forum: "Retro*: Optimizing LLMs for Reasoning-Intensive Document Retrieval"
_ICLR.cc/2026/Conference — ICLR 2026 Poster_

### Official Review · Reviewer_t3di · 2025-11-01

**Soundness:** 2
**Presentation:** 2
**Contribution:** 2
**Rating:** 6
**Confidence:** 4

**Summary:**

The paper developed Retro*, a reasoning-based re-ranking method to score the relevance of the documents for reasoning-intensive retrieval. By leveraging distillation from a teacher model, then followed by reinforcement learning with composite reward evaluation, Retro* learns to score documents using verbal reasoning from the two-stage training strategy, and works together with test-time scaling strategies (multiple rollouts). As a result, Retro scores significant performance improvement over other re-ranking baselines on BRIGHT reasoning-intensive retrieval benchmark.

**Strengths:**

1. The paper is well-written with clarity.
2. Incorporating rubrics as part of the input and having both intra- and inter- document rewards for better learning are reasonable and original ideas.
3. The empirical results are strong. The ablation studies show that the two-stage training is essential for performance improvements, with most of the gain coming from RL with composite reward.
4. It’s interesting to see from the test-time scaling experiments that having an ensemble of scores from the same reranking model offers consistent performance improvement.

**Weaknesses:**

1. While the rubrics are an important part of the contribution, having them in the model may restrict the generalizability to other reasoning-intensive tasks that are unseen.
2. The paper mentioned that the training data is based on BGE-Reasoner-Data, but it is unclear to the reader how BGE-Reasoner-Data is curated. It’s helpful to clarify that in the paper.

**Questions:**

1. The performance on Leetcode degrades significantly after reranking. I wonder if the authors have some understanding of why this is happening?
2. Why test-time scaling by average can boost the performance by so much since the aggregation is average? I can understand that without averaging, there might be more randomness, but it is a bit surprising to see that this type of ensemble can lead to performance gain.
3. Are there any potential data leakage in the experiments, especially when using the SFT model?

---

> ### Author Response · Authors · 2025-11-22
> **Rebuttal by Authors (Part 1/2)**
>
> Dear Reviewer t3di,
>
> Thank you very much for your thorough review and constructive feedback! In our response, we have carefully addressed your comments with the following discussions.
>
> > **W1**: *While the rubrics are an important part of the contribution, having them in the model may restrict the generalizability to other reasoning-intensive tasks that are unseen.*
>
> - **Our rubric is naturally generalizable by design.** It consists of two complementary components with different roles in the model's reasoning process:
>   - Relevance Definition (task-specific). This part is provided in the input and describes what relevance means for the current retrieval task. A new task simply requires providing a new definition of relevance.
>   - Scoring Criteria (task-agnostic). This part is universal across all tasks. It provides a general scope and rules for assigning relevance scores, independent of domain or task type.
> - **Experimental results on the unseen tasks confirm that Retro\* generalizes effectively.** To assess this generalizability, we evaluate Retro* and simply replace the task-specific relevance definitions on R2MED[1], a biomedical reasoning retrieval benchmark that lies outside the scientific, programming, and math domains used during training. Following ReasonRank, we adopt E5-mistral-7b-instruct as the first-stage retriever, re-rank the top-100 candidates, and report nDCG@10. All baseline results are taken directly from ReasonRank. As shown in Table 1, Retro* generalizes strongly to this unseen task, outperforming other re-ranking baselines, demonstrating the generalizability of our rubric design.
>
> Table 1: Performance on R2MED benchmark (nDCG@10), where all methods re-rank the top-100  documents retrieved by E5-mistral-7b-instruct.
>
> | E5-mistral-7b-instruct | Rank1 (7B) | Rank1 (32B) | ReasonRank (7B) | ReasonRank (32B) | Retro* (7B) | Retro* (32B) | Retro* (7B)(Mean-score @16) | Retro* (32B)(Mean-score @16) |
> | ---------------------- | ---------- | ----------- | --------------- | ---------------- | ----------- | ------------ | --------------------------- | ---------------------------- |
> | 24.0                   | 32.3       | 39.1        | 39.5            | 42.9             | 38.7        | 44.5         | 42.6                        | 46.2                         |
>
> > **W2**: *The paper mentioned that the training data is based on BGE-Reasoner-Data, but it is unclear to the reader how BGE-Reasoner-Data is curated. It’s helpful to clarify that in the paper.*
>
> - We thank the reviewer for this helpful suggestion. BGE-Reasoner-Data is an openly released dataset constructed by the BGE authors to cover all domains in the BRIGHT benchmark. For each BRIGHT task, the dataset provides synthetic queries together with retrieved documents that are annotated as relevant or irrelevant, forming positive and negative training pairs. We will add a clearer description of BGE-Reasoner-Data in the revised manuscript.
>
> > **Q1**: *The performance on Leetcode degrades significantly after reranking. I wonder if the authors have some understanding of why this is happening?*
>
> - **The performance drop on LeetCode is primarily caused by a large number of false negatives in the LeetCode corpus.** All re-ranking methods we evaluated on BRIGHT show a substantial performance drop on LeetCode compared with the first-stage retriever, suggesting that this issue stems from the dataset rather than the re-ranking method itself.
> - **The construction process of LeetCode corpus does not remove false negatives from CodeSearchNet.** BRIGHT only performs false-negative removal for documents from LeetCode, while no filtering is applied to CodeSearchNet documents. Since these documents constitute the majority of the retrieval pool, this leads to a large number of relevant examples being incorrectly labeled as irrelevant.
> - **We provide several illustrative examples in Table 2 of CodeSearchNet false negatives.**
>
> Table 2. Example false negatives in the LeetCode dataset.
>
> | query id | problem type         | one false negative document id       |
> | -------- | -------------------- | ------------------------------------ |
> | 1        | N-Queens             | leetcode/csn_python_train_119698.txt |
> | 10       | First Bad Version    | leetcode/csn_python_train_321049.txt |
> | 20       | Queue Reconstruction | leetcode/csn_python_train_3814.txt   |

---

> ### Author Response · Authors · 2025-11-22
> **Rebuttal by Authors (Part 2/2)**
>
> > **Q2**: *Why test-time scaling by average can boost the performance by so much since the aggregation is average? I can understand that without averaging, there might be more randomness, but it is a bit surprising to see that this type of ensemble can lead to performance gain.*
>
> - **Averaging smooths variability across sampled trajectories and effectively captures the model's internal consensus.** Rubric-based scoring is generative and reasoning-dependent, meaning different sampled trajectories may analyze relevance differently. For example, in the TheoremQA case (query id `TheoremQA_jianyu_xu/Ramsey_3.json`, document id `12361`), scores from 16 samples vary significantly from 40 to 87 (`[40, 40, 40, 40, 40, 40, 85, 40, 40, 60, 87, 40, 85, 85, 40, 40]`), showing that individual reasoning paths may disagree even when produced by the same model. By aggregating multiple trajectories, averaging combines these reasoning paths and captures a more robust internal consensus, consistent with prior findings on self-consistency improving reasoning performance [2].
> - **Averaging naturally provides more informative ordering without any external tie-breaking strategy.** This makes averaging a simple yet appropriate design choice for generative relevance scoring.
>
> > **Q3**: *Are there any potential data leakage in the experiments, especially when using the SFT model?*
>
> - **No data leakage in the training data.** To verify this, we perform a query-overlap analysis between BRIGHT test queries and all training queries. For each test query, we retrieve the top-20 most similar training queries using BM25 and compute a weighted Jaccard similarity. As shown in Table 3, the results indicate that all similarity scores are low, confirming that no training queries overlap with the test queries.
> - **No data leakage occurs during the SFT stage.** In the SFT stage, the student model only learns from trajectories generated by the teacher model. This process does not introduce any information from the test data, ensuring that SFT training does not compromise the integrity of the evaluation.
>
> Table 3: Maximum similarity between BRIGHT test queries and training queries, confirming no data leakage.
>
> |          | Bio. | Earth. | Econ. | Psy. | Rob. | Stack. | Sus. | Leet. | Pony | AoPS | TheoQ. | TheoT. |
> | -------- | ---- | ------ | ----- | ---- | ---- | ------ | ---- | ----- | ---- | ---- | ------ | ------ |
> | SFT Data | 0.21 | 0.23   | 0.24  | 0.20 | 0.25 | 0.22   | 0.23 | 0.31  | 0.20 | 0.27 | 0.26   | 0.27   |
> | RL Data  | 0.21 | 0.23   | 0.28  | 0.23 | 0.23 | 0.21   | 0.24 | 0.30  | 0.21 | 0.25 | 0.30   | 0.27   |
>
> [1] Li L, Zhou X, Liu Z. R2MED: A Benchmark for Reasoning-Driven Medical Retrieval[J]. arXiv preprint arXiv:2505.14558, 2025.
>
> [2] Wang X, Wei J, Schuurmans D, et al. Self-consistency improves chain of thought reasoning in language models[J]. arXiv preprint arXiv:2203.11171, 2022.

---

> > ### Comment · Reviewer_t3di · 2025-11-23
> >
> > Thanks for promptly answering the questions! Most of my concerns are now addressed. However:
> >
> > W2: I hope the authors could provide a description of BGE-Reasoner-Data and justify its usage during the rebuttal. This is important because it not only a standard practice in paper writing, but it also helps the readers to understand the similarity of data distribution between the training data and the evaluation data, also addressing my Q2.
> >
> > Q2: Could you provide semantic similarity with embedding models other than the lexical Jaccard similarity?

---

> > > ### Author Response · Authors · 2025-11-24
> > > **Rebuttal by Authors**
> > >
> > > We thank the reviewer for the helpful suggestions and the opportunity to clarify these points.
> > >
> > > > **W2**: *I hope the authors could provide a description of BGE-Reasoner-Data and justify its usage during the rebuttal. This is important because it not only a standard practice in paper writing, but it also helps the readers to understand the similarity of data distribution between the training data and the evaluation data, also addressing my Q2.*
> > >
> > > - **Description and justification of BGE-Reasoner-Data.** As much as we know, the queries in BGE-Reasoner-Data are constructed using a document-to-query generation pipeline. In this process, documents from each BRIGHT domain are used as source material to generate synthetic reasoning-intensive queries. These queries are then used to retrieve the top 100 candidate documents, which are finally annotated as positive or negative using powerful LLMs. It shares similar workflows as concurrent data curation works, such as ReasonIR and ReasonRank. Since the training queries are synthetically generated and distinct from test queries, no evaluation queries are exposed during training. We will incorporate this corresponding information into the revised manuscript and update the submission as soon as this is done.
> > >
> > > > **Q2**: *Could you provide semantic similarity with embedding models other than the lexical Jaccard similarity?*
> > >
> > > - **Semantic similarity verification beyond lexical similarity.** We additionally analyze semantic similarity using three embedding models: bge-m3[1], gte-qwen2-7b-instruct[2] and qwen3-embedding-8b[3]. For each model, we compute cosine similarity between training queries and test queries for each BRIGHT domain. As shown in Table 1, these results show consistently low semantic similarity, indicating clear discrepancy between training queries and test queries, which is consistent with our previous lexical Jaccard analysis. Together, these observations indicate that data leakage is unlikely for this curated dataset.
> > >
> > > Table 1: Semantic similarity between BRIGHT test queries and training queries.
> > >
> > > |                       | Avg. | Bio. | Earth. | Econ. | Psy. | Rob. | Stack. | Sus. | Leet. | Pony | AoPS | TheoQ. | TheoT. |
> > > | --------------------- | ---- | ---- | ------ | ----- | ---- | ---- | ------ | ---- | ----- | ---- | ---- | ------ | ------ |
> > > | bge-m3                | 0.42 | 0.38 | 0.41   | 0.42  | 0.44 | 0.44 | 0.40   | 0.43 | 0.46  | 0.51 | 0.39 | 0.39   | 0.38   |
> > > | gte-qwen2-7b-instruct | 0.28 | 0.24 | 0.26   | 0.28  | 0.26 | 0.31 | 0.23   | 0.26 | 0.33  | 0.25 | 0.29 | 0.33   | 0.31   |
> > > | qwen3-embedding-8b    | 0.17 | 0.09 | 0.13   | 0.13  | 0.14 | 0.27 | 0.12   | 0.13 | 0.18  | 0.28 | 0.21 | 0.18   | 0.18   |
> > >
> > > [1] Chen J, Xiao S, Zhang P, et al. Bge m3-embedding: Multi-lingual, multi-functionality, multi-granularity text embeddings through self-knowledge distillation[J]. arXiv preprint arXiv:2402.03216, 2024.
> > >
> > > [2] Li Z, Zhang X, Zhang Y, et al. Towards general text embeddings with multi-stage contrastive learning[J]. arXiv preprint arXiv:2308.03281, 2023.
> > >
> > > [3] Zhang Y, Li M, Long D, et al. Qwen3 Embedding: Advancing Text Embedding and Reranking Through Foundation Models[J]. arXiv preprint arXiv:2506.05176, 2025.

---

> ### Author Response · Authors · 2025-11-26
>
> Dear Reviewer t3di,
>
> I hope this message finds you well.
>
> We have just **uploaded a revised version of our paper** which incorporates your feedback. We wanted to ensure that this revision, along with our previous responses, has satisfactorily addressed all your concerns. If there are any additional points or feedback you'd like us to consider, please let us know. Your insights are invaluable to us, and we are eager to address any remaining issues to improve our work.
>
> Thank you for your time and effort in reviewing our paper.

---

### Official Review · Reviewer_d5S7 · 2025-11-01

**Soundness:** 3
**Presentation:** 3
**Contribution:** 3
**Rating:** 6
**Confidence:** 4

**Summary:**

This paper proposes Retro* a rubric-based relevance scoring mechanism for reasoning-intensive retrieval. Retro* leverage a two-stage SFT+RL training with composite intra-/inter-document rewards. The method achieves strong nDCG@10 on BRIGHT and shows consistent gains from model scale and test-time sampling.

**Strengths:**

1. Retro* uses rubric-based scoring methods, yields interpretable absolute scores rather than only relative ranks.
2. Retro* propose intra/inter rewards for RL, and the paper ablates each components.
3. The proposed methods achieve SOTA BRIGHT results.

**Weaknesses:**

1. The model uses two types of rewards (intra- and inter-document), but it remains unclear which dominates training dynamics or contributes more to final gains in run-time.
2. As rubric-guided scoring is claimed to enhance interpretability, more qualitative or case-based evidence would strengthen this claim.
3. The reliance on a strong teacher (Qwen3-235B) for SFT trajectory generation raises reproducibility and cost concerns, as smaller or open models may not reproduce the same reward quality.

**Questions:**

n/a

---

> ### Author Response · Authors · 2025-11-22
> **Rebuttal by Authors (Part 1/2)**
>
> Dear Reviewer d5S7
>
> Thank you very much for your thorough review and constructive feedback! In our response, we have carefully addressed your comments with the following discussions.
>
> > **W1**: *The model uses two types of rewards (intra- and inter-document), but it remains unclear which dominates training dynamics or contributes more to final gains in run-time.*
>
> - **Our results indicate that the intra-document reward dominates the training dynamics by enabling more effective learning, while both intra- and inter- rewards are essential and complementary.** The combination of two rewards is defined as \\(\alpha \cdot R_{\mathrm{intra}} + (1 - \alpha) \cdot R_{\mathrm{inter}}\\), and the evaluation results after training with different $\alpha$ values are presented in Table 1 (also shown in *Appendix F of our paper*).
>   - **Intra-document reward dominates learning dynamics.**  Comparing two weighted combinations, ($\alpha=0.75$, 36.6) and ($\alpha=0.25$, 36.1), we observe that assigning more weight to the intra-reward consistently leads to higher performance. Furthermore, using only the intra-reward ($\alpha=1.00$, 33.2) also outperforms using only the inter-reward ($\alpha=0.00$, 30.8), confirming that the intra-reward provides the stronger optimization signal necessary for effective training.
>   - **Both intra- and inter-document rewards are essential and complementary.** Although the intra-reward is dominant, combining intra- and inter-document rewards further improves performance. Specifically, the results in Table 1 show that any weighted combination of two rewards consistently exceeds the 33.2 of intra-reward only. This indicates that combining the rewards effectively leverages the complementary information from both signals, with performance peaking when biased toward the intra-reward.
> - **In conclusion, while the inter-reward provides necessary ranking discrimination, the intra-document reward dominates the training dynamics by stabilizing the optimization process.** This reward incentivizes the model to produce consistent trajectories, substantially reducing noise during training, enabling the model to both learn effectively and fully exploit the ranking signals provided by the inter-document reward.
>
> Table 1: Performance of models trained with different intra- and inter-document reward weightings $\alpha$.
>
> | only-SFT | +SFT+RL ($\alpha=0.00$) | +SFT+RL ($\alpha=0.25$) | +SFT+RL ($\alpha=0.50$) | +SFT+RL ($\alpha=0.75$) | +SFT+RL ($\alpha=1.00$) |
> | -------- | ----------------------- | ----------------------- | ----------------------- | ----------------------- | ----------------------- |
> | 30.1     | 30.8                    | 36.1                    | 35.8                    | 36.6                    | 33.2                    |
>
> > **W2**: *As rubric-guided scoring is claimed to enhance interpretability, more qualitative or case-based evidence would strengthen this claim.*
>
> - Retro* enhances interpretability from two perspectives:
>   - **Quantitative reliability of scores: Retro\*'s scores carry meaningful, semantically grounded information.** As shown in *Figure 4 of our paper*, positive and negative documents are clearly separated across multiple datasets. This separation emerges because the scores are defined according to rubrics (e.g., 0-20: irrelevant, 40-60: moderately relevant, 80-100: highly relevant). Consequently, a score such as 50 or 90 conveys an absolute level of relevance independent of other candidates. In contrast, RankLLaMA produces uncalibrated logits without semantic interpretation, and Rank1 reduces relevance to binary labels, which prevents any meaningful interpretation of partial relevance.
>   - **Qualitative explainability through reasoning: each score is justified by explicit reasoning.** As detailed in *Appendix K of our paper*, Retro* first analyzes the query and document and then assigns a score based on that analysis. This means every score is traceable to a rationale rather than being an unexplained numeric output.
> - In summary, Retro*'s scores are interpretable because they are both semantically grounded and reasoning-derived, rather than being relative rankings without concrete semantic meaning.

---

> ### Author Response · Authors · 2025-11-22
> **Rebuttal by Authors (Part 2/2)**
>
> > **W3**: *The reliance on a strong teacher (Qwen3-235B) for SFT trajectory generation raises reproducibility and cost concerns, as smaller or open models may not reproduce the same reward quality.*
>
> - **Reproducibility is not a major issue with modern deterministic inference.** Recent frameworks such as SGLang now support **deterministic inference**[1], which ensures that SFT trajectory generation can be reproduced reliably across different settings. This significantly mitigates reproducibility concerns even when using a large teacher model.
> - **The cost of using Qwen3-235B is practical and affordable.** Although Qwen3-235B is large, it is a MoE model, and inference activates only **22B active parameters**. In our setup, we deploy Qwen3-235B on 8 NVIDIA H100 GPUs. Generating trajectories for 12,000 training samples, with 8 trajectories per sample, takes **less than 6 hours** in total. We believe this overhead is affordable and well within the budget of academic or industrial compute environments.
> - **Our method can achieve competitive re-ranking performance even without using any SFT trajectories from teacher model.** In the only-RL setting with the composite reward, the model already achieves an average nDCG@10 of 35.1, outperforming other baselines of similar model size. The strong teacher mainly provides initial reasoning capability to stabilize and accelerate exploration during RL, but **is not required** for the method to reach strong re-ranking performance.
>
> [1] Towards Deterministic Inference in SGLang and Reproducible RL Training

---

> ### Author Response · Authors · 2025-11-26
>
> Dear Reviewer d5S7,
>
> I hope this message finds you well.
>
> We have just **uploaded a revised version of our paper** which incorporates your feedback. We wanted to ensure that this revision, along with our previous responses, has satisfactorily addressed all your concerns. If there are any additional points or feedback you'd like us to consider, please let us know. Your insights are invaluable to us, and we are eager to address any remaining issues to improve our work.
>
> Thank you for your time and effort in reviewing our paper.

---

### Official Review · Reviewer_9eA5 · 2025-11-03

**Soundness:** 3
**Presentation:** 3
**Contribution:** 3
**Rating:** 2
**Confidence:** 4

**Summary:**

This paper presents Retro*, a new method for training rerankers for reasoning intensive queries. Retro* is first trained with both SFT on traces from a large powerful model and then later trained with reinforcement learning. For the later stage, the authors design new reward functions specifically for improving reranking performance.

**Strengths:**

The strengths of the paper are as follows:
- The RL training rewards proposed in this paper are novel and are distinct from other baseline methods.
- The parallelism and inference time analysis is interesting.

**Weaknesses:**

The weaknesses of the paper are:
- Lately, there has been a lot of debate whether reasoning is helpful for reranking. Is the reasoning trajectory actually necessary or can model be trained directly to just produce the score. The authors should ablate whether the reasoning trajectory are helpful and if so how helpful they are.
- For the retro* results in table 1, the authors use first-stage candidates from BGE-Reasoner-Embed. What is the first stage retriever for the other baseline models in main table? If the first stage retriever is different then it is hard to compare numbers.
- Along similar line, the authors should clarify that the baseline models are not build from the same backbone. For instance, RankLLaMA and RankZephyr are built from older llama models.
- The performance gains seem limited when making comparisons in the same setting. For instance, ReasonIR+QwenRerank obtains a score of 36.9 on bright and this is with no additional reranker training (the qwen model instruct model is used as is). From table 2, we see that Retro* obtains a scores of 37.4 after SFT and RL training.
- ReasonIR is mentioned in the paper but is not used as a baseline. I understand that ReasonIR is a retriever but the ReasonIR paper also presents numbers for ReasonIR+a zero-shot qwen based reranker.
- Rubric-based relevance scoring is not novel. Table 10 of the Rank1 paper shows that data-specific prompts were used for BEIR and the non-stackoverflow BRIGHT subsets. ReasonIR did something similar as well. Also the name "rubric-based scoring" is a little mis-leading because it implied there are multiple rubric elements, but the rubric is just a definition of similarity for each task.
- Minor issues: There are some typos in the paper. E.g. "while positive documents are assigned low scores", "with zero RL achieves a higher score of 35.1."

**Questions:**

Here are some questions:
- I think there is a typo on line 214. There are four chosen trajectories and not just one, right?
- For the only-SFT ablation how many training samples were used?

---

> ### Author Response · Authors · 2025-11-22
> **Rebuttal by Authors (Part 1/3)**
>
> Dear Reviewer 9eA5,
>
> Thank you very much for your thorough review and constructive feedback! In our response, we have carefully addressed your comments with the following discussions.
>
> > **W1:** *Lately, there has been a lot of debate whether reasoning is helpful for reranking. Is the reasoning trajectory actually necessary or can model be trained directly to just produce the score. The authors should ablate whether the reasoning trajectory are helpful and if so how helpful they are.*
>
> - **Our experiments demonstrate that reasoning is necessary for re-ranking and yields substantial performance gains.** To evaluate this directly, we train a variant of Retro* on the same training data and using the same RL algorithm, but without any reasoning trajectory or rubric-guided analysis. We focus this comparison on the RL stage to avoid introducing biases from the teacher model used in SFT. This setup keeps the learning signal identical, isolating the effect of reasoning. As shown in Table 1 below, removing reasoning leads to substantial degradation across all datasets.
> - **Our results indicate that rubric-guided reasoning improves the fidelity of relevance estimation under RL optimization.** Structured reasoning decomposes the task and provides intermediate reasoning that clarifies how different aspects of relevance contribute to a judgment. Without this intermediate reasoning supervision, the model only learns relevance from a scalar score. This weak signal results in unreliable scoring, poor generalization on unseen domains, and potential risk of overfitting.
>
> Table 1. Performance comparison on the BRIGHT benchmark, showing that reasoning-enhanced re-ranking significantly outperforms models trained without reasoning.
>
> | Model               | Avg. | Bio. | Earth. | Econ. | Psy. | Rob. | Stack. | Sus. | Leet. | Pony | AoPS | TheoQ. | TheoT. |
> | ------------------- | ---- | ---- | ------ | ----- | ---- | ---- | ------ | ---- | ----- | ---- | ---- | ------ | ------ |
> | No Reason (only RL) | 24.8 | 38.4 | 40.2   | 22.3  | 32.4 | 20.5 | 25.6   | 26.0 | 19.6  | 16.9 | 4.7  | 17.7   | 33.8   |
> | Reason (only RL)    | 35.1 | 56.7 | 52.9   | 33.2  | 45.1 | 28.1 | 32.5   | 35.8 | 21.2  | 36.3 | 8.6  | 29.8   | 41.0   |
>
> > **W2:** *For the retro\* results in table 1, the authors use first-stage candidates from BGE-Reasoner-Embed. What is the first stage retriever for the other baseline models in main table? If the first stage retriever is different then it is hard to compare numbers.*
>
> - **All results in Table 1 of our paper are directly comparable, as all baselines re-rank** **the same first-stage candidates** (the top-100 documents retrieved by BGE-Reasoner-Embed). This experimental setup is already presented in our paper (Evaluation Settings section, caption of Table 1).
>
> > **W3:** *Along similar line, the authors should clarify that the baseline models are not build from the same backbone. For instance, RankLLaMA and RankZephyr are built from older llama models.*
>
> - We thank the reviewer's observation. To address it clearly, we organize our response as follows.
> - **All reasoning-enhanced baselines in our paper are built on the same backbone family, Qwen2.5-Instruct.** In contrast, the non-reasoning baselines RankLLaMA and RankZephyr are built upon **Llama-2** and **Mistral-v0.1**, respectively. The revised version will explicitly clarify these backbone differences in the *Baselines and Backbone Models* section.
> - **Retro\* consistently outperforms RankLLaMA and RankZephyr even when using the same backbone models.** To further compare with these non-reasoning baselines, we additionally train Retro* on the same backbones used by RankLLaMA and RankZephyr. As shown in Table 2, Retro* with Llama-2 reaches 25.4 (+5.7) over RankLLaMA, and with Mistral-v0.1 7B reaches 27.8 (+7.0) over RankZephyr , confirming that the performance gains come from our proposed method rather than backbone capacity.
>
> Table 2. Retro* outperforms RankLLaMA and RankZephyr when using the same backbone models.
>
> | Re-ranking on BGE-Reasoner-Embed | RankLLaMA (7B) | RankZephyr (7B) | Retro* (Llama-2 7B) | Retro* (Mistral-v0.1 7B) |
> | -------------------------------- | -------------- | --------------- | ------------------- | ------------------------ |
> | w/o retriever scores                   | 19.7           | 20.8            | 25.4                | 27.8                     |

---

> ### Author Response · Authors · 2025-11-22
> **Rebuttal by Authors (Part 2/3)**
>
> > **W4:** *The performance gains seem limited when making comparisons in the same setting. For instance, ReasonIR+QwenRerank obtains a score of 36.9 on bright and this is with no additional reranker training (the qwen model instruct model is used as is). From table 2, we see that Retro\* obtains a scores of 37.4 after SFT and RL training.*
> >
> > **W5:** *ReasonIR is mentioned in the paper but is not used as a baseline. I understand that ReasonIR is a retriever but the ReasonIR paper also presents numbers for ReasonIR+a zero-shot qwen based reranker.*
>
> - **ReasonIR+QwenRerank uses a different evaluation setting, and therefore cannot be directly compared to ours.** QwenRerank computes its final score by combining retriever and re-ranker scores using $0.5 \times S_{reranker} + 0.5 \times S_{retriever}$, where both scores are normalized (as stated in Section E.2 of the ReasonIR paper). In contrast, our evaluation uses only the re-ranker score $S_{reranker}$ as the final ranking signal, so that the results reflect the reranker's own capability without influence from the retriever.
> - **Under the same evaluation setting, Retro\* significantly outperforms QwenRerank.** As shown in Table 3, without retriever fusion, QwenRerank achieves only 30.3 (reported in Table 7 of the ReasonIR paper), while Retro* reaches 37.4 under the same setup. Under the same fusion setting, ReasonIR+Retro* also consistently yields higher performance than ReasonIR+QwenRerank. These results indicate that most of the gains attributed to QwenRerank largely come from the strong retriever ReasonIR, whereas Retro* consistently achieves higher performance purely from its re-ranking capability.
>
> Table 3. Retro* consistently outperforms QwenRerank under both pure re-ranking and score-fusion settings.
>
> | Re-ranking on ReasonIR | QwenRerank (32B) | Retro* (7B) | Retro* (7B) (Mean-Score@16) | Retro* (32B) | Retro* (32B) (Mean-Score@16) |
> | ---------------------- | ---------------- | ----------- | --------------------------- | ------------ | ---------------------------- |
> | w/o retriever scores   | 30.3             | 36.8        | 38.4                        | 37.4         | 39.5                         |
> | w/ retriever scores    | 36.9             | 38.4        | 39.0                        | 39.6         | 40.2                         |
>
> > **W6**: *Rubric-based relevance scoring is not novel. Table 10 of the Rank1 paper shows that data-specific prompts were used for BEIR and the non-stackoverflow BRIGHT subsets. ReasonIR did something similar as well. Also the name "rubric-based scoring" is a little mis-leading because it implied there are multiple rubric elements, but the rubric is just a definition of similarity for each task.*
>
> - Although the mentioned papers introduced similar task-specific prompts or definitions, our method introduces the following fundamental distinctions:
>   - **Higher flexibility and finer granularity**. Our relevance rubric explicitly specifies both the relevance definition and a fine-grained scoring criteria with salient landmarks (e.g., 0, 20, 40, 60, 80, 100). This design enables the model to flexibly assign nuanced scores across a 0-100 with interpretable meaning. In contrast, the mentioned methods restrict the model to coarse and fixed classification choices, e.g., binary prediction (yes/no) as used by Rank1.
>   - **Support for effective reinforcement.** The flexibility of our rubric-based relevance scoring allows us to derive intra-document reward signals, which is crucial for effective RL optimization (as shown in Table 3 of our paper). In contrast, when predictions are restricted to discrete and coarse categories, the scores often tie, making it difficult or even impossible to compute reliable reward signals. As a result, prior methods cannot stably benefit from reinforcement learning in principle.
>   - **Effective optimization through reinforcement.** Our experiments show that reinforcement learning is essential for making rubric-based scoring effective. Retro* is the first method to systematically optimize rubric-based scoring, rather than relying on simple prompting. While existing methods such as QwenRerank do not provide corresponding learning algorithm and therefore leave the rubric-based relevance scoring underexplored.

---

> ### Author Response · Authors · 2025-11-22
> **Rebuttal by Authors (Part 3/3)**
>
> > **W7**: *Minor issues: There are some typos in the paper. E.g. "while positive documents are assigned low scores", "with zero RL achieves a higher score of 35.1."*
>
> - We thank the reviewer for pointing out the writing issues in our paper. We will revise the paper accordingly based on these suggestions.
>
> > **Q1**: *I think there is a typo on line 214. There are four chosen trajectories and not just one, right?*
>
> - **Only one trajectory is selected for each query-document pair.** For each pair, we generate multiple raw trajectories, compute their average score, and then choose the trajectory whose score is closest to this average to construct the SFT training data. The mention of "4-tuples" in the text refers only to the content of each data instance $(q, d, y, s)$, not to selecting four trajectories. We will revise the presentation in the manuscript to explicitly clarify that only one trajectory is selected for each query-document pair.
>
> > **Q2**: *For the only-SFT ablation how many training samples were used?*
>
> - **The only-SFT ablation uses 12,000 training samples, the same as in the main SFT stage (described in the** ***Training Data*** **section of our paper).** All experimental settings in the ablation study remain the same to the main experiments, except for the component being ablated. We will explicitly clarify this in the ablation study section of the manuscript.

---

> ### Author Response · Authors · 2025-11-26
>
> Dear Reviewer 9eA5,
>
> I hope this message finds you well.
>
> We have just **uploaded a revised version of our paper** which incorporates your feedback. We wanted to ensure that this revision, along with our previous responses, has satisfactorily addressed all your concerns. If there are any additional points or feedback you'd like us to consider, please let us know. Your insights are invaluable to us, and we are eager to address any remaining issues to improve our work.
>
> Thank you for your time and effort in reviewing our paper.

---

> ### Author Response · Authors · 2025-11-28
>
> Dear Reviewer 9eA5,
>
> Thank you once again for your valuable comments on our submission. **As the discussion phase is approaching its end**, we would like to kindly confirm whether we have sufficiently addressed all of your concerns (or at least part of them). Should there be any remaining questions or areas requiring further clarification, please do not hesitate to let us know. If you are satisfied with our responses, we would greatly appreciate your consideration in adjusting the evaluation scores accordingly.
>
> We sincerely look forward to your feedback.

---

### Author Response · Authors · 2025-11-30
**General Responses by Authors**

Dear Program Chairs, Senior Area Chairs, Area Chairs and Reviewers,

We sincerely thank you for your thorough review and constructive feedback! During the rebuttal period, we conduct extensive additional analyses and experiments to address the reviewers' concerns and questions. Below is a summary of our major response efforts.

- **Summary for Reviewer 9eA5.**

  Reviewer 9eA5 expressed three major concerns about our paper, to which we offer the following clarifications.

  - **Question about the necessity of reasoning for re-ranking (W1).** We conduct additional experiments that clearly confirm the necessity of reasoning for effective re-ranking. Particularly, following the reviewer's suggestion, we train a variant of Retro* that directly outputs a score without any reasoning trajectory, which results in a substantial performance drop across all BRIGHT domains. The corresponding analyses and results have been incorporated into the revised paper.
  - **Question about the fairness of the experiments (W2, W3, W4, W5).** These concerns arise from the **reviewer's misinterpretations** of our experimental settings. In our experiments, all baselines re-rank exactly the same first-stage candidates (W2). Moreover, except for the two non-reasoning baselines, all reasoning-enhanced baselines are built upon the same Qwen2.5 backbone family (W3). We also clarify that our evaluation uses only re-ranker scores, whereas the reported results of QwenRerank are based on combining retriever and re-ranker scores. Under the same evaluation settings, Retro* achieves substantially higher performance than QwenRerank (W4, W5). In other words, our method **fairly outperforms the baselines** across all situations. We have made these points clear in the response, and have empahsized corresponding discussions in the revised paper.
  - **Question about the novelty of methodology (W6).** Our approach introduces higher flexibility and finer granularity by explicitly specifying both the relevance definition and detailed scoring criteria. It supports effective reinforcement by enabling reliable reward signals and is the first to systematically optimize rubric-based scoring through reinforcement learning, making the method practically usable and substantially more powerful than prior prompt-based approaches.

- **Summary for Reviewer d5S7 and Reviewer t3di.**

  Both Reviewer d5S7 and Reviewer t3di **recognize the main contributions of our work**, including achieving SOTA performance, the novelty and effectiveness of our reinforcement learning algorithm, and the effectiveness of our test-time scaling approach. We next summarize their specific concerns and our corresponding responses.

  - **The concerns of Reviewer d5S7.** We clarify that the intra-document reward dominates training dynamics while both intra- and inter-document rewards are complementary, supporting effective learning (W1). We also demonstrate that rubric-guided scoring enhances interpretability by producing semantically grounded, well-calibrated relevance scores and by providing explicit reasoning that justifies each score (W2). Finally, we show that SFT with a strong teacher is reproducible and cost-efficient, while not being strictly required for achieving strong performance (W3).
  - **The concerns of Reviewer t3di.** We show that the rubric naturally generalizes to unseen domains by evaluating on R2MED (W1). We provide a detailed description of BGE-Reasoner-Data, explain its construction process, and justify its usage (W2). We also address the reviewer's remaining questions regarding LeetCode performance, test-time averaging, and potential data leakage (Q1, Q2, Q3).

In the revised manuscript, we have incorporated the clarifications and supplementary results discussed during the rebuttal. We believe these updates further strengthen the paper and resolve the raised concerns.

---

### Meta-Review · Area_Chair_w8CZ · 2025-12-27

**Summary:**

The authors propose a RL-based method for training a reranker. Reviewer d5S7 and reviewer t3di were generally positive; reviewer 9eA5 noted several concerns around novelty and experimental design that the authors responded to in their rebuttal. The additional experiments during the rebuttal were persuasive. Overall, the proposed method is straightforward but still a new and useful contribution.

**Reviewer Concerns:**

Reviewer 9eA5 had concerns around experimental design (e.g., making sure that the first-stage candidates were the same for all methods, and comparisons vs prior work in QwenRerank), but these were addressed in the rebuttal. The other reviewers' concerns were also addressed.

**Reviewer Scores:**

d5S7: 6 -> 7
t3di: 6 -> 7
9eA5: 2 -> 4

---

### Decision · Program_Chairs · 2026-01-26

Accept (Poster)